



# Peatland bog pedogenesis is reflected in unsaturated hydraulic properties

Tobias K. D. Weber, Sascha C. Iden, Wolfgang Durner

*Soil Science and Soil Physics Division, Institute of Geoecology, Technische Universität Braunschweig, Langer Kamp 19c,*
*38106 Braunschweig, Germany.*

*Correspondence to*: Tobias KD Weber (to.weber@tu-bs.de)

**Abstract.** In ombrotrophic peatlands, the moisture content of the acrotelm (vadoze zone) controls oxygen diffusion rates, redox state, and the turnover of organic matter. Thus, variably saturated flow processes determine whether peatlands act as sinks or sources of atmospheric carbon, and modelling these processes is crucial to assess effects of changed environmental

conditions on the future development of these ecosystems. We show that the Richards equation can be used to accurately describe the moisture dynamics under evaporative conditions in variably saturated peat soil, encompassing the transition from the topmost living moss layer to the decomposed peat as part of the vadose zone. Soil hydraulic properties (SHP) were identified by inverse simulation of evaporation experiments on samples from the entire acrotelm. To obtain consistent descriptions of the observations, the traditional van Genuchten-Mualem model was extended to account for non-capillary water

storage and flow. We found that the SHP of the uppermost moss layer reflect a pore-size distribution (PSD) that combines three distinct pore systems of the Sphagnum moss. For deeper samples, acrotelm pedogenesis changes the shape of the SHPs due to the collapse of inter-plant pores and an infill with smaller particles. This leads to gradually more homogeneous and bi-modal PSDs with increasing depth, which in turn can serve as a proxy for increasing state of pedogenesis in peatlands. From this, we derive a nomenclature and size classification for the pore spaces of Sphagnum mosses and define inter-, intra-, and

inner-plant pore spaces, with effective pore diameters of $>°300°\mu m$, $300–30°\mu m$ and $30–10°\mu m$, respectively.

## 1 Introduction

*Sphagnum* moss is the dominant and keystone species in peatland development (Kuhry and Vitt, 1996). As ecosystem engineer, *Sphagnum* moss (Jones et al., 1994) induces vertical growth of peat bogs leading to subsequent ombrotrophication (Rydin and Jeglum, 2016), and has led to considerable terrestrial carbon accumulation during the Holocene (Frolking and Roulet, 2007).

The ability of *Sphagnum* spp. to photosynthesize critically depends on the water regime and is limited to the topmost centimetres of the bog profile where growth occurs (Clymo, 1973). Under field conditions, soil hydraulic properties (SHP) control the water regime in the topmost, growing part of the peat profile and are thus a critical factor for moss growth and survival (Hájek and Beckett, 2008). The importance of capillary, film and vapour flow for upward water fluxes in moss and





peat has been emphasized by Hayward and Clymo (1982) and Price et al. (2009) while SHPs accounting for these processes have only recently been identified by Weber et al. (2017a) for a limited number of samples.

In natural and undisturbed peat bog ecosystems there is a transition from living moss near the surface to heavily decomposed moss peat at greater depths (Clymo and Hayward, 1982, Limpens et al., 2008, Morris et al., 2011), with a concomitant change

in SHPs (e.g. Quinton et al., 2008; Price et al., 2008; McCarter and Price, 2012, 2014; Morris et al., 2015; Weber et al., 2017a). As *Sphagnum* decomposes, the fibrous material is biochemically broken up into smaller solid fragments (Rezanezhad et al., 2016). This change in physical structure reduces the pore sizes by a collapse of large pores and fine material filling the voids of the organic soil matrix. An increase in overburden leads to vertical shrinkage due to compression and a resulting increase in dry bulk density with depth (Clymo, 1978; Johnson et al., 1990; Price, 2003). The resulting strong decrease in saturated

hydraulic conductivity, $K_s$ (L T$^{-1}$) (Boelter, 1969; Ingram, 1978; Quinton et al., 2008) is positively correlated with the state of decomposition as shown by Ingram (1978), Clymo (1970, 1984), Clymo and Hayward (1982), Hayward and Clymo (1982), and Morris et al. (2015). Exceptions from this rule have been observed in cases where pipe flow (Holden, 2005), fire disturbances (Sherwood et al, 2013), and rapid climate change resulting in changes in vegetation and subsequent peat deposition history (Rydin and Jeglum, 2016, Hedwall et al., 2017) occur.

In soil hydrology, the Richards equation is widely used as process model for simulating temporally and spatially variable water contents and fluxes in soils. Recently, Weber et al. (2017a) demonstrated that the Richards equation is an adequate process model for unsaturated water flow under evaporative conditions even for the uppermost layer of a bog, where the living *Sphagnum* moss can be treated as part of the soil. The solution of the Richards equation requires the parametrization of SHPs. For systems with depth-dependent SHPs like peat bogs, one can assume either a homogeneous system and identify effective

SHP, or parametrize the SHP as depth-dependent functions (Durner et al., 2008, Vereecken, 2007). In peatland hydrology, the saturated hydraulic conductivity $K_s$ is the most frequently measured hydraulic parameter for different depths. However, saturated conductivity is not necessarily a good predictor for the unsaturated hydraulic conductivity curve. While the correlation of the saturated hydraulic properties, i.e. $K_s$ and porosity with depth has been well established, corresponding changes of the SHP, namely the water retention characteristic (WRC) and particularly the unsaturated hydraulic conductivity

curve (HCC) with depth have neither been extensively investigated nor been treated in numerical simulations.

In ecohydrological modelling of peatland mires containing *Sphagnum*, unimodal expressions of the WRC still dominate (e.g. Dimitrov et al., 2011, Sulman et al., 2012; Wu and Blodau, 2013; Mezbahuddin et al. 2016). However, the investigation of the WRC of organic soils has revealed bimodal pore size distributions which has been found for different, but limited pressure head ranges (Rezanezhad et al., 2009, Quinton et al. 2008, Quinton et al. 2009, Rezanezhad et al., 2010, Kettridge and Binley,

2011, and Rezanezhad et al., 2016), and have so far been rarely expressed in terms of effective SHPs over the full pressure head range.

Weber et al. (2017a) applied the highly sensitive method of inverse modelling evaporation experiments performed on *Sphagnum* moss and peat over a wide moisture range to parameterise SHPs. The WRCs over the pressure head range from -1500 cm to 0 revealed a trimodal PSD and clearly showed the necessity to account for adsorbed water in drying soil. Samples



from two depths of the acrotelm were analysed and the identified SHPs differed systematically and in a complex manner for the two depths. The HCC could not be described by simply scaling the saturated hydraulic conductivity, and showed the relevance of capillary and film components for water flow in organic soils. The trimodal nature of the PSD of the sample with living *Sphagnum* moss was linked to three conceptual pore spaces: pores between plant individuals (inter-plant pores), pores

between branches and leaves of individual plants (intra-plant pores), and plant-internal water stored in hyaline cells (inner-plant pores). The deeper sample with decomposed moss showed a bimodal PSD with pores associated to the organic soil matrix and pores to the hyaline cells and its skeletal remnants (Fig. 1).

From this we state three hypotheses for this study. First, we expect a gradual shift in the characteristic pore sizes of the inter-plant and inner-plant pores with depth, due to increasing compaction and decomposition of the living moss material. Second,

we expect a gradual disappearance of the distinct modalities (particularly the one that indicates the inter-plant pores) and an associated widening of the effective pore-size distribution with depth. Finally, we hypothesize that the variability of replicate samples from the same depth decreases with increasing depth, which is associated with the disappearance of the inter-plant pores. The last hypothesis is based on the assumption that samples become more homogeneous due to pedogenic processes such as biochemical degradation and compaction, and that the typical representative elementary volume for the different pore-

classes differ, being largest for the inter-plant pores.

Correspondingly, the research aims of this study are i) to investigate in detail the gradual change of SHPs with depth, encompassing the transition from living moss to peat, ii) to analyse and quantify the spatial variability in SHPs with depth, and iii) to provide a unifying framework to distinguish between different pore systems in *Sphagnum* moss and peat which is based on the PSD and distinct pressure head ranges. To achieve these goals, a series of 28 laboratory evaporation experiments

was conducted on undisturbed samples from the topmost 40 cm of a bog profile. The measured data were evaluated by inverse modelling using the Richards equation as process model with the objective to identify SHPs for living moss and *Sphagnum*-derived peat. The results are presented in terms of depth-averaged SHPs, PSDs, and air entry pressure heads of the individual pore systems.

## 2 Materials and Methods

### 2.1 Sampling site and sample preparation

The samples were collected at a soli-ombrotrophic peat bog, the Odersprungmoor, Harz Mountains, Central Germany (UTM 32U 608000 mE 5737000 mN; 800 to 821 m a.s.l.). The bog formed in a valley position in the SW-NE direction and on a saddle with an average slope of 3 % (Fig. 2; Jensen, 1990). The long-term average temperature is 6.8°C and annual precipitation is 1270 mm with a high inter-annual variance. The open bog is almost elliptical in shape with the longer axes oriented in SE-

NW direction and has an area of 16.9 ha which is surrounded by spruce forest. The predominant bog vegetation consists of *Sphagnum magellanicum* and *Sphagnum rubellum* with interdispersed *Eriophorum angustifolium*. Further extensive vegetation mapping was carried out by Baumann (2009). Figure 2 provides a detailed spatial reference and an overview on the peat types





in the bog, the minerotrophic water flow into the bog on the North-Western flank, the watershed divide and the sampling location. Broder and Biester (2015) provide information on geochemical composition of the substrate and pore waters. Weber et al. (2017b) and Gerling et al. (2017) give a detailed account of the carbon, water and energy fluxes of the site during the vegetation period of 2013, a year with exceptionally dry summer time conditions in comparison to the 136 year long term
record measured by the meteorological station of the German Weather Service (DWD) nearby Braunlage.

In the acrotelm, a profile characterization with depth is possible according to a weak grouping: the first 15 cm are composed of living moss, in 15-30 cm dead plant remnants dominate with some parts of the plants still visible, and the 30-40 cm samples contain the dark coloured decomposed moss peat. After the first 15 cm of living moss, a continuous increase in the state of decomposition is measurable (Broder and Biester, 2015). From the profile, 28 cylindrical samples of 250 cm³ volume and 5
cm height were obtained from frozen large sampling blocks using a drill rig (Quinton et al., 2009; Weber et al.; 2017a). We will refer to the samples by the respective mid sampling depths, i.e. 2.5 (5x), 7.5 (6x), 12.5 (3x), 17.5 (3x), 22.5 (3x), 32.5 (4x), and 37.5 (4x) cm (number of replicates in brackets). Freezing is considered to be a negligible source of error (Branham and Strack, 2014). A comprehensive description of sampling protocol, sample treatment, and methodology is provided by Weber et al. (2017a).

**2.2 Identification of soil hydraulic properties**

Initially, the samples were saturated using de-aired and de-ionized water over a period of 48 h. The saturated hydraulic conductivity, $K_s$ (cm d⁻¹) of all samples was determined by the falling head-method (KSAT device version 1.4; UMS GmbH, München, Germany). Subsequently, transient evaporation experiments were carried out (Wendroth et al., 1993, Schwärzel et al. 2006). The experimental details are described by Weber et al. (2017a). The measured data were evaluated by inverse
modelling using the Hydrus-1D code to solve the Richards equation numerically (Šimůnek et al., 2016). Model parameters were identified by minimizing a weighted least-squares objective function (OF), which contained the time series of pressure heads in two depths (1.25 and 3.75 cm) and the sample's average water content. Six different parametrizations of the SHP which can be categorized into two main groups were tested (Weber et al. 2017a). The first model group is based on the widely used van Genuchten-Mualem model (van Genuchten, 1980), which conceptualises all pore water to be contained and conducted
in capillaries. We used the unimodal (VGM1), bimodal (VGM2), and trimodal (VGM3) van Genuchten model to parametrize the SHP. The uni- and multimodal van Genuchten models are used widely in soil hydrology and the equations are therefore not repeated here in detail, but can be found, e.g., in Priesack and Durner (2006). The second model group consists of the Peters-Durner-Iden (PDI) model of the SHP. The model is physically more comprehensive in that it ensures zero water content at oven-dryness, and accounts for water flow in completely-filled capillaries, partly-filled capillaries, adsorbed water films,
and isothermal vapour diffusion. We used the parametrization derived by Peters (2013), in the modified form published by Iden and Durner (2014) and Peters (2014). The PDI retention curve model has an identical number of free parameters as the VGM. For the conductivity curve, two additional parameters are required which express i) the relative contribution of a non-capillary conductivity to the total unsaturated conductivity, and ii) the steepness of the decrease of the non-capillary





conductivity curve with increasing suction. Similar to the VGM models, the PDI models were applied in uni, bi- and trimodal form (PDI equations are given in the Annex).

The data from the evaporation experiments on the 28 samples were evaluated by inverse modelling using all six models of the SHP (VGM1, VGM2, VGM3, PDI1, PDI2, PDI3). In the parameter optimization, the saturated conductivity parameter, $K_s$,

was fixed to the measured value for each individual sample. This resulted in six sets of estimated model parameters, corresponding SHPs, and model performance criteria per sample. Model selection was based on the Akaike information criterion corrected for small sample size, AICc, as defined in Ye et al. (2008) and discussed in Weber et al. (2017a).

### 2.3 Calculation of mean soil hydraulic properties

After identifying the best parametrization of the SHPs for each sample depth based on the AICc, the SHPs of the replicates

were averaged to obtain representative, mean SHPs for each depth. Due to the nonlinear dependence of the SHPs on the model parameters, average SHPs were not calculated by averaging parameters. Since the matric potential ranges across orders of magnitude in drying soil, we will use, for convenience sake, the pF unit in the remainder of this article. It is defined as the common logarithm of the negative pressure head $h$(cm), i.e., pF = $\log_{10}$(-$h$) (-). The averaging was done by first generating 197 equidistant support points on the on the pF interval [-3, 6.8]. Subsequently, point values of the WRC and HCC were

calculated at the support points for each replicate based on its respective best model. The resulting datasets of the replicates were binned into one dataset and a new set of average PDI3 parameters was estimated by nonlinear least-squares fitting. Fitting was done sequentially, first the average WRC model parameters were estimated and in a second step, the retention parameters were kept constant and the parameters of the HCC were estimated using log conductivity data in the objective function. The sequential parameter estimation circumvents the need to weight the data groups of $\theta(h)$ and $\log_{10} K(h)$. An example of the

averaging method is illustrated in Fig. 3. The iterative minimization of least squares was done with the Differential Evolution algorithm (Price et al. 2006) as implemented in the R package DEoptim (Mullen et al.; 2011).

### 2.4 Pore-size distributions and classification of pore sizes

According to the Young-Laplace equation (Jury and Horton, 2004) the pressure head at which a capillary tube drains is inversely proportional to the equivalent effective diameter, $d_{eff}$ (L), of that capillary which is calculated by:

$$d_{eff} = \frac{4\,\gamma\,cos\,\beta}{|h|\rho_w g} \qquad (1)$$

where $\gamma$ is the surface tension (M T$^{-2}$) of water at 20°C, $h$ is pressure head (L), $\rho_w$ is the density of water (M L$^{-3}$), $g$ is the gravitational acceleration (L T$^{-1}$), and $\beta$ (-) the contact angle between water and the solids which we here set to 0° (Valat et al., 1991), i.e. complete wettability is assumed during monotonic drying (Goetz and Price, 2015).

The PSD function, $f(h)$, was defined as change of capillary saturation with log pressure head (Durner, 1994):

$$f(h) = -\frac{dS_c}{d[log_{10}(-h)]} = -\log(10)\,|h|\frac{dS_c}{dh} \qquad (2)$$





where $S_c$ is the capillary saturation function defined by Eqs. (A2) and (A3) in the appendix. In multimodal pore systems, this can be applied to the capillary saturation functions of each pore system $S_{c,i}$ (-), which define the individual pore systems that are superimposed in the effective PSD.

The analysis of the systematic changes in SHPs with depth is based on Eq. (1). We characterise the depth evolution of $d_{eff}$ by the evolution of the median pore diameter, $d_{50}$, and the ratio of the 75$^{th}$ to the 25$^{th}$ quantile, $d_{75}/d_{25}$ of the pore size distributions (Eq. 2). This is in analogy to characterisations of particle size distributions.

A size classification of the individual pore domains was based on pressure heads at which air entry occurs to the different pore systems defining the PSD. In a first step, air-entry pressure heads, $h_{ae,i}$, were determined for each superimposed effective PSD of each depth. From the seven determined values each for the first, second, and third PSD of the depth averaged SHPs, as described in section 2.3, a profile average for each modality was computed to determine the air entry.

In case of the van Genuchten saturation model used in the PDI models, the air entry pressure is usually approximated as the reciprocal of parameter $\alpha$ (Assouline et al., 1998). However, the steepness of the WRC is controlled by shape parameter $n$ and only for relatively large values of $n$ is the air-entry pressure indeed approximately equal to $\alpha^{-1}$. Therefore, we define the air-entry pressure of a pore system as the pressure head where $S_{c,i} = 0.95$. For the weighted saturation function of the van Genuchten model, the equation to calculate the approximate air-entry value pressure of the $i$th pore system is given by the equation

$$h_{ae,i} = \alpha_i{}^{-1} * \left(0.95^{-1/m_i} - 1\right)^{1/n_i} \qquad (3)$$

. This equation can equally be used for the PDI model. Equation (3) shows that the air-entry pressure depends not only on shape parameter $\alpha$ but additionally on parameter $n$ which characterizes the width of the PSD.

As mentioned above, three pressure heads are calculated delimiting the three domains. Towards the dryer end, the pressure head limit between domains corresponds to the adjacent domains air entry pressure. However, only residual water remains when pressures heads are within the range of the last domain. To address this this, a pressure head delimiting the last capillary domain was determined at which the dominant compound of water transport changes from capillary to non-capillary flow, where the capillary and non-capillary conductivity curves intersect. At this point, the capillary conductivity drops so rapidly that it gives a sharp delimitation between the two water storage and conductivity domains.

## 3 Results and Discussion

### 3.1 Model performance

The statistical metrics for all 168 inverse simulations (28 samples times 6 models) are listed in Table S1. These encompass the root mean squared weighted errors (RMSE$_w$) which characterize model performance, the AICc which enable model comparisons, and the minimum objective function values. The values for the best performing model for each sample are summarised in Table 1. According to the AICc, the PDI3 model performed best for 24 out of the 28 samples. For the remaining





samples, the VGM3 model performed best in three cases and the PDI2 in one. Thus, the PDI3 leads to the best model prediction in the majority of replicates regardless of sampling depths. We recall that this is achieved by estimating one additional conductivity curve parameter in the PDI compared to the VGM models with the same modality. This finding challenges the de facto standard usage of the VGM1 and VGM2 models for parametrizing the SHPs of Sphagnum moss and

peat and extends the theoretical considerations, results, and conclusions in Weber et al. (2017a) to multiple depths in the bog profile. Note that the use of an incorrect model, despite lower number of parameters, often leads to very high parameter uncertainties like in Dettman and Bechtold (2016). As a consequence, the averaging of the SHP among the replicates was based on the PDI3 model for all depths.

### 3.2 Soil hydraulic properties and pore-size distributions of all samples

Figure 4 presents the WRCs (left), HCCs (middle), and PSDs (right) of all measured samples, grouped for the seven sampling depths (denominated A to G, from top to bottom). A comparison of the SHPs at the different depths reveals three main trends. First, the variability between the replicates decreases with increasing depth, i.e. SHPs of the replicates become more similar. Second, for all depths a comparable modality of relatively small pores is discernible, which starts to drain at about pF = 2 (-100 cm). Superimposed to this pore system, there is a second and third modality indicating larger pore systems for the near-

surface samples, with the third modality reflecting a pore domain close to water saturation. With increasing depth the variability in the two larger pore domains decreases considerably. Third, the HCCs for all depths appear amazingly similar in the more unsaturated moisture range (pF > 2), but diverge towards saturation. One possible reason for the scatter for pF < 2 is the insensitivity of evaporation experiments to changes in the $K(h)$ function in wet soil. This insensitivity occurs if the evaporation rate is on the order of $K(h)$ or smaller, leading to small gradients in pressure head which cannot be resolved by tensiometers

(Peters and Durner, 2008). A more reliable determination of the HCC close to water saturation could be obtained by the tension disk method (Klute and Dirckson, 1986; McCarter et al. 2017), multi-step outflow (Durner and Iden, 2010), or multi-step flux measurements (Weller et al., 2011). However, these methods have their own challenges when applied to organic soils and the latter two have, to the best of our knowledge, not been applied to obtain SHPs for organic media so far, although Qi et al. (2011) used MSO experiments to determine the WRC.

### 3.3 Change of pore-system characteristics with depth

To better identify the pore system changes with depth, we compare mean SHPs for all depths in Fig 5. The trends with depth which have been observed for the individual samples, now become much clearer. The seven mean WRCs (Fig 5, left) show an increased water holding capacity with increasing depth across a wide pressure head range, from saturation up to pF = 3.2 (a pressure head of -1500 cm). This finding is supported by literature data for conditions near water-saturation (Price et al., 2008;

Price and Whittington 2010; McCarter and Price, 2012; Goetz and Price; 2015). Concurrently, the HCCs show a pronounced decrease of the saturated conductivity and systematically higher unsaturated conductivities with increasing depth (Fig 5, right), but show a crossing point of the conductivity functions at around pF = 1.





From an ecophysiological point of view, it is interesting to analyse the relationship between SHPs and the resilience against desiccation. In fact, all samples have a high water capacity around pF = 2. During meteorological conditions favouring prolonged periods of drying, the decrease in pressure head is slowed down by the release of this water, and low pressure heads leading to desiccation and cessation of photosynthesis (Hájek and Beckett, 2008) are reached only slowly. Parallel to the WRC,

the slope of the HCC around pF = 2 decreases as water drains from hyaline cells and their skeleton material, and hydraulic conductivity remains at values of approximately $10^{-2}$ cm d$^{-1}$ until approx. pF = 2.5, where it drops at all depths rapidly below $10^{-4}$ cm d$^{-1}$. The relatively high values of hydraulic conductivity in the pressure head range until pF = 2.5 ensures an upward flow of water which contributes to desiccation tolerance under field conditions.

We further observe that the WRCs shifts with depth from a trimodal pore-size distribution to a bimodal one with a distinct

expression of the two remaining pore systems. This is nicely recognizable from the plots of the PSDs in Fig. 6. To calculate meaningful pore-size parameters, we separated the fine pore system, characterized by the capillary saturation function with the parameters $w_3$, $a_3$ and $n_3$, from the system of larger pores, defined by the superposition of the two saturation functions with parameters $w_2$, $a_2$, $n_2$ and $w_1$, $a_1$, and $n_1$, and calculated the median diameters of the two pore systems for each depth. These median diameters are indicated in Fig. 6 by the dashed vertical lines. From visual inspection, and confirmed by the

values of the fitted parameters of the third pore subsystem ( Table 2), it becomes obvious that the pore-system of the finer pore remains virtually unchanged with depth, encompassing about 15 to 20 % of the total pore space, $w_3(\theta_s - \theta_r)$. The distinct peak of this pores system is at an identical position at pF = 2.3 for all depths (Fig. 6). Also, the width of this fine pore system shows no change with depth. In contrast, the median equivalent diameter, $d_{50}$ (L), from the superimposed first and second modality drifts gradually from larger to finer pores (Fig. 6).

To further carve out the depth-dependency of key characteristics of the pore systems, we show the depth dependence of saturated conductivity and two characteristic pore-size parameters in Fig. 7. The near log-linear decrease in measured saturated hydraulic conductivity $K_s$ (Fig.7, a) is in agreement with established trends in these ecosystems (Morris et al., 2015). The shift in the median pore size of the larger pore system indicates the strongest transition between 10 and 25 cm (Fig.7, b). The width of the associated pore system, expressed by the ratio of the first and third quartile of the PSD, $d_{75}/d_{25}$ (-), shows a distinct

and continuous narrowing with depth (Fig.7, c).

As a side note, we like to comment on an interesting aspect of using the different parametrizations of the HCC. We found that the tortuosity/connectivity parameter $\tau$ in this study is positive for each individual sample parameterisation with the PDI3 (Table S1). The same is true for the calculated mean functions at all depths (Table 2). A positive value of $\tau$ is in agreement with its conceptual meaning (Mualem, 1976; Peters et al., 2011) and has also been found by Weber et al. (2017a). This contrasts

reports from the literature (Price et al., 2008; Whittington and Price, 2010) and our findings with van Genuchten type functions, where $\tau$ is often negative (Table S1). Obviously, the use of a structurally more correct model of the SHPs that accounts for partially filled capillaries, prevents parameter $\tau$ from becoming negative. Conversely, if an inadequate model of the SHPs is applied, $\tau$ has a tendency to become increasingly small and even negative because it induces the HCC to be less steep in the in the medium to dry range. Although this effect may be welcome to describe measurements, it is obtained by a physically



implausible value of the shape parameter (Hoffman-Riem et al., 1999). Another important feature of the identified values of $\tau$ is their relative small variation with depth.

The gradual change of PSDs with depth summarized by Fig. 6 can be explained by the origin of the samples and the pedogenic processes they have undergone, i.e. biochemical degradation breaking up fibres into smaller pieces (e.g. Rezanezhad et al. 2016), compaction (Price and Schlotzhauer, 1999), the shrinkage and swelling of bog soils (Schlotzhauer and Price, 1999; Price et al., 2003; Price, 2003; Glaser et al., 2004) and freeze-thaw processes (Meiers et al., 2013). These findings are corroborated by a positive relationship between depth and advance in decomposition at this site (Broder and Biester, 2015). Obviously, these processes lead to a homogenisation of the organic material, a disappearance of the largest pores, and thus to a smaller spatial variability of the SHPs in greater depths. While in the upper 15 cm the living moss reveals a large natural phenotypic variability over orders of centimetres, the dead plant remnants at 15-30 cm have already been subject to some compaction and the inter-plant pores are filled with smaller particles. The most decomposed samples from 30-40 cm, which lie beneath the water table most of the time, are the most homogeneous and have completely lost the distinct trimodality in the PSD, which was evident for the samples from near the surface. The deepest samples are also the most homogeneous in colour and solid particle sizes. It is worthwhile to note that the PSDs are basically bimodal but this does not contradict the superiority of using a trimodal parametrization (see section 3.1 and Table 1) as indicated by the AICc. The superposition of PSDs can improve the overall description, even if they do not represent two distinct pore systems with different characteristic pore sizes. This was also discussed for the bimodal van Genuchten model by Zurmühl and Durner (1998) and for organic soils by Weber et al. (2017a).

**4 Proposal of a pore size classification for *Sphagnum* moss and peat**

The results of our study suggest a pore-size classification that is based on the three different pore domains characterised by the air-entry pressures (Eq. 3; Table 3). This classification relates the PSD to different physical components of moss and peat and hereby also directly links the SHPs to pedogenesis. The resulting averaged air entry pressures for the first, second and third pore domain are given as PF 0, 1, 2, respectively, and in the following we give a detailed description of the pore water domains and the residual water. (Fig. 8, Table 4). I-III are based on capillary water storage and IV is based on non-capillary (adsorptive) water storage.

The **first domain (I)** reaches from saturation to a pressure head of approx. -10 cm (pF = 1), which is marked by the mean $\bar{h}_{ae,2}$, corresponding to effective pore diameters of > 300 μm. At this point the HCCs of different depths converge in the sense that for pF > 1 the scatter is significantly reduced and the individual curves run in parallel with a range of approximately one order of magnitude (Fig. 5Figure 5). This pressure head corresponds to a state where the highly heterogeneous *inter-plant pores* (living and slightly decayed *Sphagnum*) have drained (Fig. 1).

The **second domain (II)** extends from pF = 1 to pF = 2 corresponding to effective pore diameters between 300 μm-30 μm. It comprises the intra plant pore space in the sampling depths 2.5 cm – 27.5 cm where macroscopic living and undecomposed



moss fragments exists. In the remaining sampling depths decomposition is advanced but the skeletal structures containing the hyaline cells remains (Fig. 1); correspondingly the water in the second domain is here stored in the outer skeletal spaces of the peat (Rezanezhad et al., 2016).

Finally, the **third pore domain (II)** is represented by intact and decayed hyaline cells with effective pore radii of 30 μm-

10 μm. As shown in Fig. 1, the hyaline cells can still have intact cell membranes, but appear to decay more rapidly than the structures bearing them. In fact, the distances between two structural elements (skeleton) making up the hyaline cells is, effectively, marginally larger than the hyaline cell opening.

Thus, in the absence of macropores, two size classes can be differentiated in decomposed *Sphagnum* peat; the *outer plant matrix* pores >°30°μm, and the *inner plant matrix* pores with effective pore radii 30–10°μm (Table 4). The above mentioned

hyaline cells and their skeletat structures are drained by pF = 2.5 which is the pressure head where the **fourth domain (IV)** starts. At these pressure of pF > 2.5, the water conductance can no longer be described by capillary flow theory, a direct result of evaporating water in the flow paths no longer being replenished.

In line with these definitions, a pressure head delimitation of pore water into an active (*inter- and intra-plant and inter-plant matrix pore space*) in an inactive porosity (*inner-plant and inner plant matrix*) at a pressure head of $h = -100$ cm is suggested.

**5 Conclusions**

Based on measured data from transient evaporation experiments and a robust evaluation by inverse modelling using the Richards equation as a process model, we identified the SHPs for 28 samples from an acrotelm of an ombrotrophic, *Sphagnum* moss dominated bog. To parameterize the SHPs, we compared six different models. The successful description of the observed data was possible by a) considering three modalities representing the underlying PSD and b) including the contribution of non-

capillary water in the SHP models. Combined, this results in an encompassing parameterization across the full moisture range. With the continuous information from fully parameterised models, available from the presented work, greatly extends the level of detail in comparison to previous works.

We averaged the SHPs for each of the seven sampled depths and found, compared to insights from biological studies that these mean hydraulic properties reflect the physiological nature of the pore size domains in the living moss carpets of peatbogs.

Larger heterogeneities in identified SHP between depth replicates could be shown for samples from the living moss, however, this effect reduced considerably with increasing depth and a concomitant reduction of PSDs from tri- to bimodal. Both effects, the homogenisation and reduction in modality, can be explained by pedogenic processes.

A unifying nomenclature to describe and report results from research on the vadose zone of ombrotrophic peatlands is proposed. With this, the pore spaces in Sphagnum moss and peat can be classified according to size ranges of the effective

pore diameters. These size ranges are derived from an analysis on the shape of SHPs and underlying PSDs from the uppermost 40 cm of the acrotelm. In living Sphagnum moss, the three pore spaces are thus proposed to be referred to as *inter-, intra-,* and *inner-plant pore spaces*, with effective pore diameters of >°300°μm, 300–30°μm and 30–10°μm, respectively. For deeper





samples, the pedogenesis of the acrotelm has had a considerable impact on the shape of the SHPs. First, the collapse of the inter-plant pores and their filling with smaller particles led to more homogeneous and to gradually bi-modal PSDs with increasing depth. Much of the hyaline cell structure and skeleton remains less affected by compaction and is resilient to decomposition. These remaining larger pores with effective pore diameters of >°300°µm and fine pores 30–10°µm correspond

to more to the plant matrix than the fully visible plant and were thus described as *outer plant matrix* pores and the *inner plant matrix*, respectively.

Following these delimitations and with depth continuous differentiation between the larger pores and inner plant pores, we conclude that from a soil hydrological perspective, the water storage domain consist of an active and inactive porosity delimited at a pressure head of $h = -100$ cm. The proposed domain classification was inferred from a soil hydrological modelling

approach and with the discussion of the physiological meaning, we associate them to measurements and reflections in the seminal biological studies by Hayward and Clymo (1982), Lewis (1988), and van Breemen (1995).

Future research should include testing the Richards equation under various boundary conditions to verify that the identified SHPs can be used for predictive purposes under different boundary conditions. Additionally, the research should be conducted to establish if proxies such as bulk density, C/N rations, and/or vegetation structure can be used to predict SHPs accurately, as

is often done for saturated conductivity. Lastly, scaling approaches are expected to help describe the SHPs of bog profiles continuously.

**Code and Data availability**

Codes for and data can be made available upon request.

**Supplement**

Table S 1: Model performance criteria and parameters for all models and all depths.

**Appendix A: PDI Model Equations**

The WRC of the PDI model family (Peters, 2013, Iden and Durner, 2014, Peters, 2014) is given by the equation

$$\theta(h) = (\theta_s - \theta_r)\, S_c(h) + \theta_r S_{nc}(h) \qquad (A1)$$

where $\theta$ (-) is volumetric water content, $h$ (cm) is pressure head, and $S_c(h)$ (-) and $S_{nc}(h)$ (-) are the saturation functions for capillary and non-capillary water, respectively. $S_c(h)$ (-) is defined as:

$$S_c(h) = \frac{\Gamma(h) - \Gamma_0}{1 - \Gamma_0} \qquad (A2)$$

where $\Gamma_0 = \Gamma(h_0)$, $h_0$ (cm) is the pressure head at oven-dryness, and $\Gamma(h)$ is the multimodal van Genuchten saturation function (Durner, 1994)





$$\Gamma(h) = \sum_{i=1}^{k} \Gamma_i(h) = \sum_{i=1}^{k} w_i \left[1 + (-\alpha_i h)^{n_i}\right]^{-m_i} \tag{A3}$$

with shape parameters $\alpha_i$ (cm$^{-1}$) and $n_i$, the constraint $m_i = 1 - 1/n_i$, the number of modes or pore systems, $k$ (-), and weights $w_i$ which sum to unity. The non-capillary saturation function is given as

$$S_{nc}(h) = 1 + \frac{1}{x_a + x_0}\left\{x - x_a + \ln\left[1 + exp\left(\frac{x_a - x}{b}\right)\right]\right\} \tag{A4}$$

where $x = log_{10}(-h)$, $x_a = log_{10}(-h_a)$, $x_0 = log_{10}(-h_0)$, and $h_a = 1/\alpha_1$. Parameter $b$ is a smoothing parameter for which empirical equations are given in Iden and Durner (2014). The HCC is defined as

$$K(h) = (1 - \omega)K_s K_{rc}(S_c) + \omega K_s K_{rnc}(S_{nc}) + K_v(h) \tag{A5}$$

5 where $\omega$ (-) is the relative proportion of water flow in films and corners, $K_{rc}(h)$ (-) is relative hydraulic conductivity caused by flow in completely-filled capillaries, $K_{rnc}(h)$ (-) is relative hydraulic conductivity caused by flow in films and corners (incompletely filled capillaries), and $K_v(h)$ is the effective hydraulic conductivity for isothermal vapour diffusion. The equation for $K_v(h)$ can be found in Saito et al., (2006) and the closed-form equations for the two liquid conductivities are (Peters, 2013; Peters, 2014)

$$K_{rc}(S_c) = S_c^{\tau}\left[1 - \left(\frac{\sum_{i=1}^{k} w_i \alpha_i \left(1 - \Gamma_i^{1/m_i}\right)^{m_i}}{\sum_{i=1}^{k} w_i \alpha_i \left(1 - \Gamma_{i,0}^{1/m_i}\right)^{m_i}}\right)^m\right]^2 \tag{A6}$$

10 and

$$K_{rnc}(S_{nc}) = \left(\frac{h_0}{h_a}\right)^{a(1-S_{nc})} \tag{A7}$$

where $\Gamma_{i,0} = \Gamma_i(h_0)$ and $a = 1.5$ is an empirical parameter for film flow (Tokunaga, 2009).

**Author contributions**

TW designed the study, the experiments and carried them out. SI developed the model code and jointly performed the simulations with TW. TW prepared the manuscript with contributions from all co-authors.

15 **Competing interests**

The authors declare that they have no conflict of interest.

**Acknowledgements**

We wish to express our gratitude to the Niedersächsische Technische Hochschule for grant TD 2.1.4 Top-Down-Project GeoFluxes and for the research scholarship awarded to TKD Weber by the German Academic Exchange Service. Further, we





extend our thankfulness to Dr. Benedikt Scharnagl and Dr. Efstathios Diamantopolous for fruitful discussions, to Dr. Kison of the National Park Harz for granting site access, as well as to our esteemed fellow Geoecologists Daniela Reineke, Lennart Rolfes, and Matthias Spieckerman for their diligent work in the field and lab. For the images we wish to thank Dr. Reuven Simhayov for providing and Christina Smeaton for creating the scanning electron microscope image, and to Bent Johnsen for

the drawing of *Sphagnum magellanicum* in Fig. 1 (© Department of Biology, University of Copenhagen),

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





**Figures**

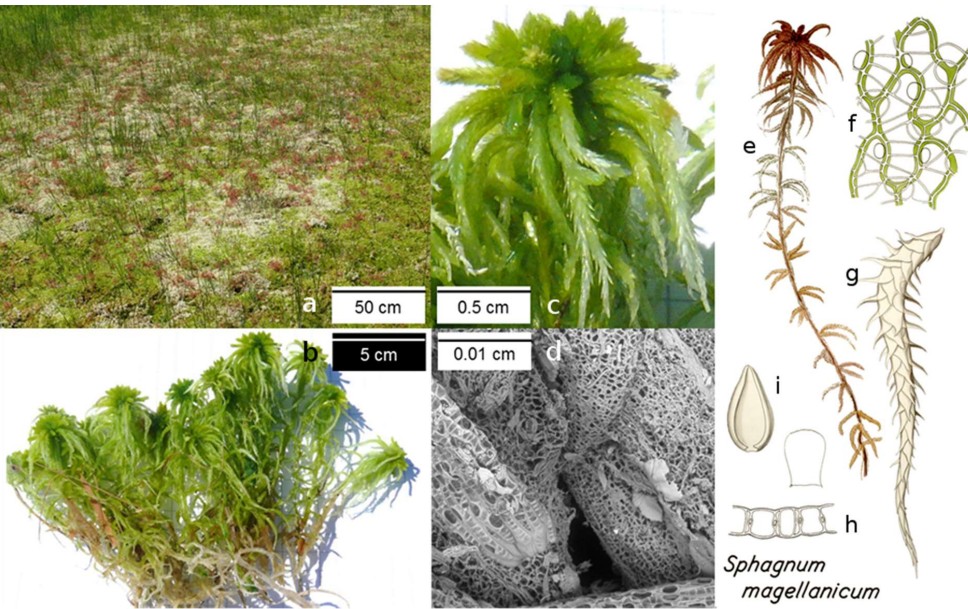

Figure 1: Sphagnum moss structures and soil pore sizes. a) Sphagnum lawn with visible bleaching due to desiccation of the capitula (in German language Sphagnum is also referred to as 'Bleichmoos', which translates to 'bleaching moss'), b) sampled and slightly spread out Sphagnum individuals with visible inter-connectedness of branches, c) close up of the capitula with pending branches and leaves, d) scanning electroscope microscopy image of Sphagnum leaves on a branch with visible dark circles as the opening to the hyaline cells (by courtesy of Reuven Simhayov). Right: drawing of Sphagnum magellanicum (by courtesy of Bent Johnsen. © Department of Biology, University of Copenhagen) with, e) individual plant, f) branch surface with skeletal structure in green, g) branch with leaves, h) cross section of a leaf with large hyaline cells, and i) individual leaf.





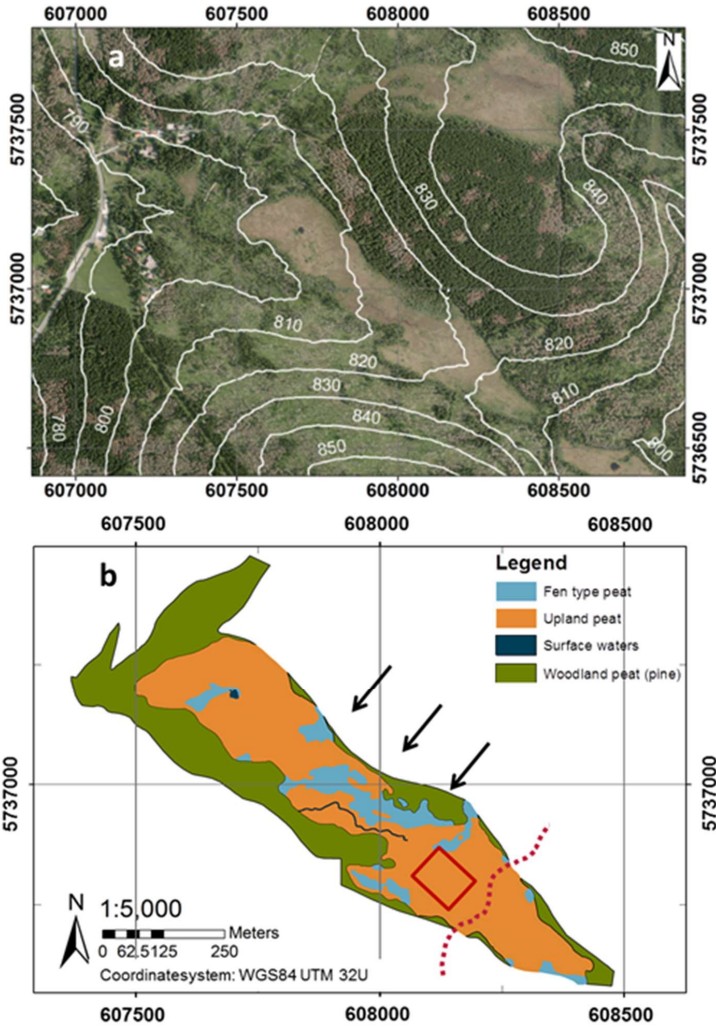

Figure 2: a) Satellite image and elevation contours of the Odersprungmoor, b) vegetation mapping and characterisation of the peatland, black arrows indicate minerotrophic water influx into the peatland resulting in higher proportion of fen type vegetation, the red dashed line is the approximate watershed divide, and the red box marks the sampling area.





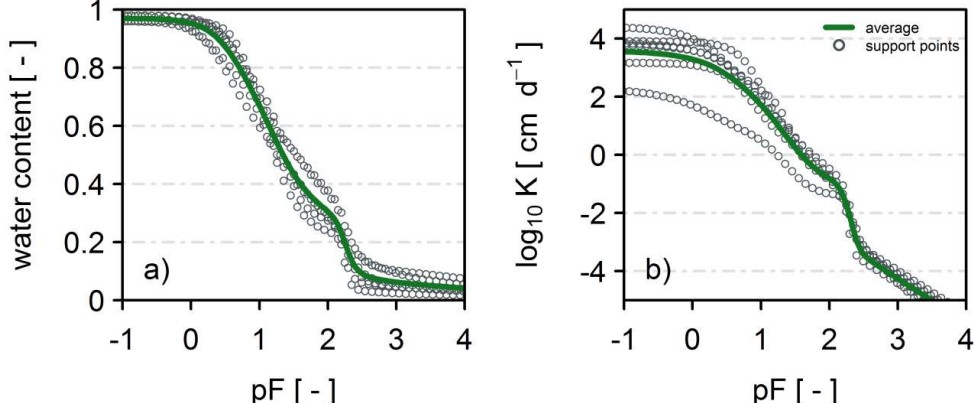

Figure 3: Example of the averaging process for (a) water retention curve and (b) hydraulic conductivity curves, shown for depth 7.5 cm (five replicates). Despite the variability of the individual curves, the key structural features are conserved.



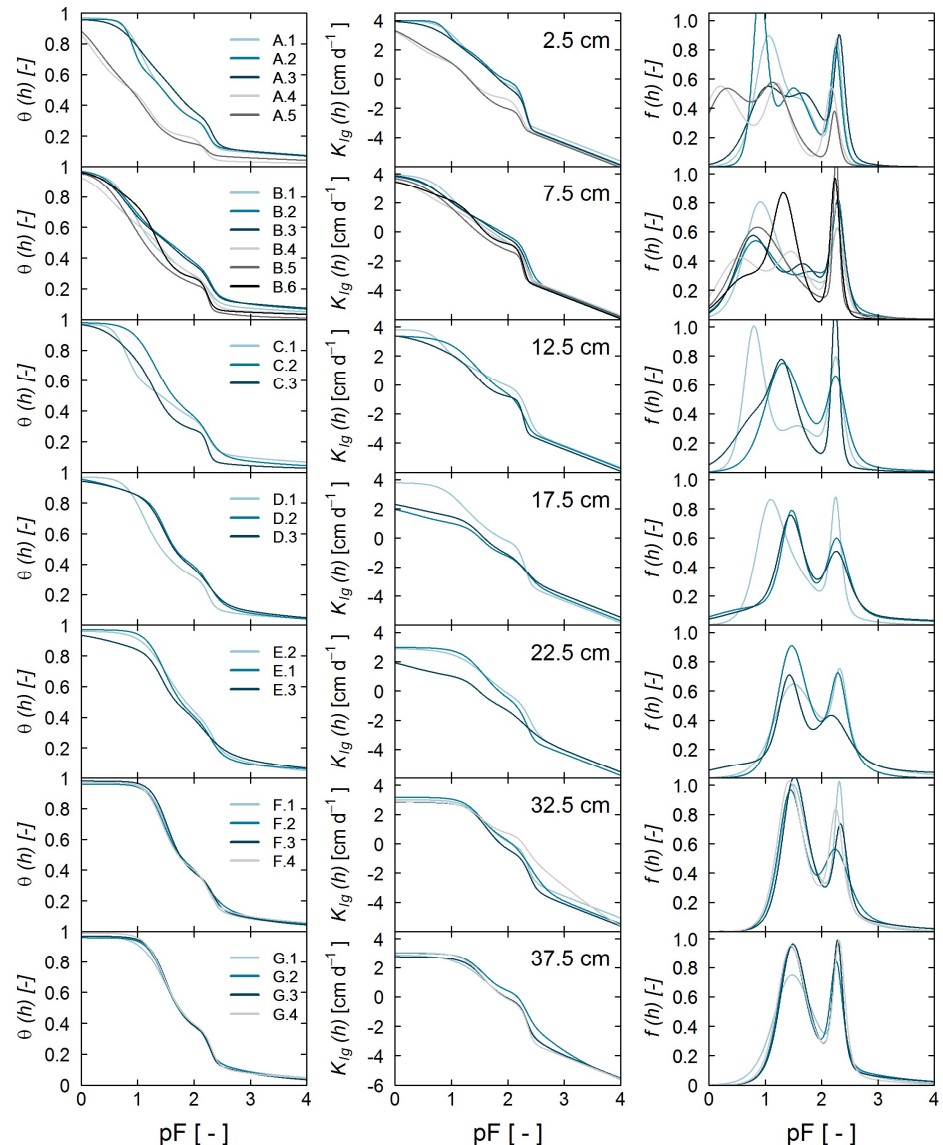

Figure 4: Left: Water retention curves, middle: hydraulic conductivity curves, and right: Pore-size distributions (of all 28 samples, $K_{lg}$ refers to the common logarithm of the conductivity value).





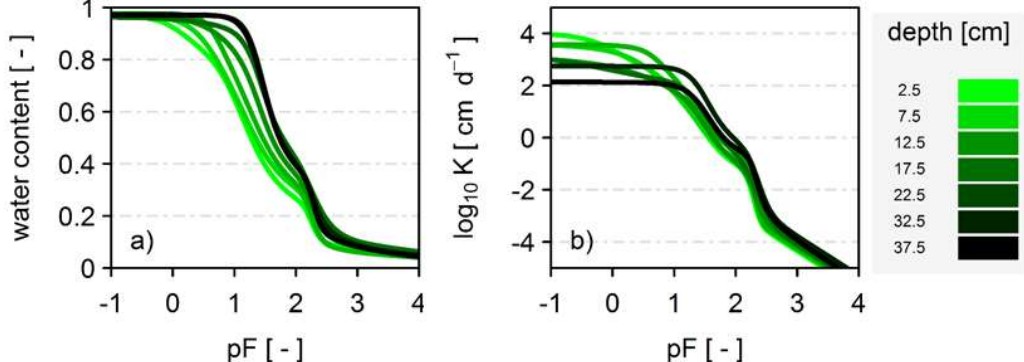

Figure 5: Mean (a) retention and (b) conductivity curves for the depths 2.5 to 37.5 cm in 5 cm increments.





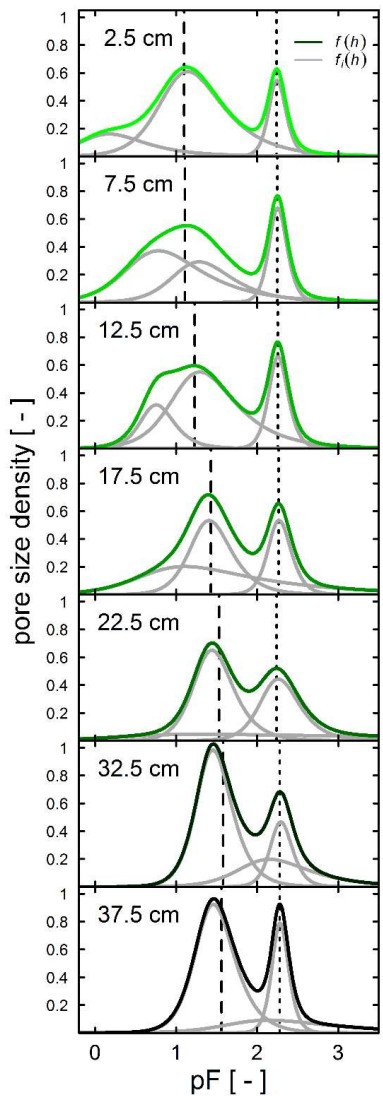

Figure 6: Continuous pore size distributions calculated from Eq. 4 per depth showing the overlap of the three pore spaces. The dashed line represents the median (d50) of the first and second pore domain which positively correlates with increasing depth. The dotted line marks the modus of the third modality which remains constant regardless of depth.





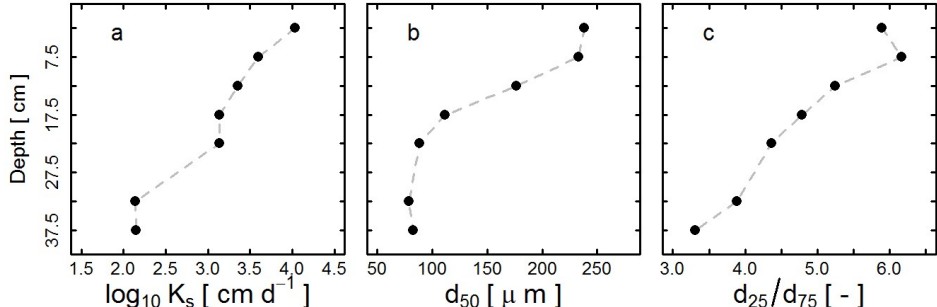

Figure 7: (a) Trends of key pore systems characteristics with depth: (a) near exponential decrease in $K_s$ with depth, (b) decrease of characteristic pore diameter with depth, expressed as median pore diameter $d_{50}$ of the second modality (c) reduction of the width of the pores-size distribution with depth, expressed by $d_{75}/d_{25}$.

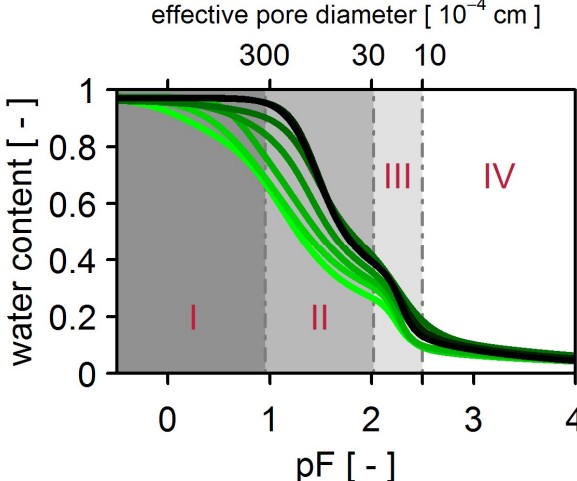

Figure 8: Four pore domains represented in the soil water retention curve. The roman letters I-III denote different capillary water domains for which a detailed explanation is given in the text and values are presented in Table 4. The region denoted by IV is non-capillary (residual) water.





**Tables**

Table 1: Statistical evaluation results of the inverse parameter estimation for 31 samples[1] of eight mid sampling depths, the objective function value (OFN)[3] value is the weighted sum of squared errors, i.e. the scalar value to be minimised during optimisation (Eq. 20 in Weber et al, (2016)), the root mean square weighted errors (RMSW$_E$[4,5]) of the matric potential

5 measurements for the heights 0.75L and 0.25L, and median values of 4.3 cm and 3.0 cm, respectively, the corrected Akaike Information criterion (AICc[6]), the difference in AICc between best and second best model (ΔAICc[7]), and best model based on the AICc, VGM3 and PDI represent the trimodal van Genuchten Mualem model and the modified Peters variant..

| Sample Name[1] | Mid sampling depth[2] | OF Value[3] | RMSE$_w$[4] 0.75 | RMSE$_w$[5] 0.25 | AICc[6] | ΔAICc[7] | Best Model[8] |
|---|---|---|---|---|---|---|---|
| [ - ] | [cm] | (x 10$^4$) | [cm] | [cm] | [ - ] | [ - ] | [ - ] |
| A.1 | 2.5 | 0.82 | 5.1 | 2.4 | 2348 | 417 | PDI3 |
| A.2 | 2.5 | 1.60 | 7.1 | 2.7 | 2734 | 795 | PDI3 |
| A.3 | 2.5 | 0.58 | 6.5 | 2.1 | 2688 | 394 | PDI3 |
| A.4 | 2.5 | 1.03 | 8.2 | 5.1 | 3518 | 321 | PDI3 |
| A.5 | 2.5 | 0.78 | 6.8 | 4.5 | 3051 | 432 | PDI3 |
| B.1 | 7.5 | 2.03 | 3.8 | 2.9 | 1766 | 806 | PDI3 |
| B.2 | 7.5 | 1.51 | 4.9 | 2.3 | 2156 | 954 | PDI3 |
| B.3 | 7.5 | 1.05 | 6.6 | 3.0 | 2600 | 413 | PDI3 |
| B.4 | 7.5 | 0.91 | 7.4 | 4.2 | 2595 | 782 | PDI3 |
| B.5 | 7.5 | 0.61 | 8.4 | 3.9 | 3692 | 350 | PDI3 |
| B.6 | 7.5 | 1.41 | 4.3 | 3.7 | 3408 | 1566 | VGM3 |
| C.1 | 12.5 | 1.17 | 4.6 | 2.3 | 1856 | 642 | PDI3 |
| C.2 | 12.5 | 1.29 | 8.6 | 8.0 | 5562 | 27 | PDI2 |
| C.3 | 12.5 | 1.12 | 5.0 | 4.6 | 3594 | 797 | PDI3 |
| D.1 | 17.5 | 0.95 | 2.6 | 4.9 | 2362 | 185 | PDI3 |
| D.2 | 17.5 | 2.31 | 3.5 | 3.8 | 3209 | 1727 | PDI3 |
| D.3 | 17.5 | 2.06 | 1.9 | 2.4 | 1727 | 2788 | PDI3 |
| E.1 | 22.5 | 1.81 | 2.2 | 2.3 | 1724 | 1182 | PDI3 |
| E.2 | 22.5 | 2.45 | 4.1 | 2.1 | 2418 | 957 | PDI3 |
| E.3 | 22.5 | 3.99 | 2.6 | 3.7 | 2684 | 1654 | PDI3 |
| F.1 | 32.5 | 3.69 | 3.0 | 2.9 | 1966 | 958 | VGM3 |
| F.2 | 32.5 | 2.52 | 2.1 | 3.9 | 2347 | 186 | PDI3 |
| F.3 | 32.5 | 1.26 | 4.3 | 1.5 | 2102 | 447 | PDI3 |
| F.4 | 32.5 | 0.95 | 4.8 | 3.5 | 2166 | 94 | VGM3 |
| G.1 | 37.5 | 2.79 | 3.8 | 4.3 | 2640 | 653 | PDI3 |
| G.2 | 37.5 | 1.90 | 3.0 | 3.2 | 2162 | 470 | PDI3 |
| G.3 | 37.5 | 0.54 | 2.6 | 2.7 | 1724 | 1424 | PDI3 |
| G.4 | 37.5 | 0.38 | 4.6 | 2.1 | 2445 | 156 | PDI3 |





Table 2: PDI3-model parameter values of the depth averaged effective soil hydraulic properties, $\theta_r$, residual water content, $\theta_s$, saturated water content, $\alpha_i$ and $n_i$ are air entry and shape parameters of the three super imposed capillary bundles, $w_i$ is the weighting coefficient of the capillary bundles, $K_s$ the saturated and $\omega$ the non-capillary conductivity, and $\tau$ the strictly positive tortuosity parameter. The
5   common logarithm to the base 10 is denoted by $lg$. A detailed model description is given by Weber et al. 2016. The parameters of all individual samples can be found in the Supplementary Information. Bulk density can be calculated from knowledge of $\theta_s$ which can be treated as porosity.

| Depth | $\theta_r$ | $\theta_s$ | $\alpha_1$ | $n_1$ | $w_1$ | $\alpha_2$ | $n_2$ | $w_2$ | $\alpha_3$ | $n_3$ | $lg\,K_s$ | $\tau$ | $lg\,\omega$ |
|---|---|---|---|---|---|---|---|---|---|---|---|---|---|
| (cm) | (-) | (-) | (cm$^{-1}$) | (-) | (-) | (cm$^{-1}$) | (-) | (-) | (cm$^{-1}$) | (-) | (cm d$^{-1}$) | (-) | (-) |
| 2.5 | 0.12 | 0.97 | 0.96 | 2.0 | 0.18 | 0.1 | 2.1 | 0.66 | 0.006 | 6.4 | 4.0 | 0.6 | -5.9 |
| 7.5 | 0.09 | 0.97 | 0.26 | 1.8 | 0.50 | 0.07 | 2.1 | 0.30 | 0.006 | 6.2 | 3.6 | 0.6 | -4.2 |
| 12.5 | 0.10 | 0.98 | 0.19 | 3.3 | 0.20 | 0.07 | 2.0 | 0.60 | 0.006 | 7.5 | 3.4 | 1 | -4.2 |
| 17.5 | 0.06 | 0.97 | 0.18 | 1.4 | 0.42 | 0.05 | 2.9 | 0.37 | 0.006 | 4.9 | 3.1 | 1.0 | -5.9 |
| 22.5 | 0.04 | 0.96 | 0.44 | 1.1 | 0.24 | 0.04 | 2.9 | 0.46 | 0.006 | 3.1 | 2.8 | 1.7 | -6.0 |
| 32.5 | 0.09 | 0.97 | 0.04 | 3.4 | 0.55 | 0.02 | 1.7 | 0.25 | 0.005 | 5.3 | 2.1 | 0.2 | -5.6 |
| 37.5 | 0.06 | 0.97 | 0.04 | 3.0 | 0.61 | 0.02 | 1.5 | 0.17 | 0.005 | 6.6 | 2.1 | 0.3 | -5.7 |





Table 3: Air entry pressures of the three pore systems of the effective SHPs are presented for all sampling depths (left column) and averages as the arithmetic mean to delimit the individual pore domains. (*) As previously established, the depths 32.5 and 37.5 show no macro porosity and are excluded from calculating the mean and standard deviation for the average $\bar{h}_{ae,1}$-value. The mean $\bar{h}_{ae,2}$-value is calculated from the the $h_{ae,2}$-values of the sampling depths 2.5-27.5 cm, and the two $h_{ae,1}$-values at depths 32.5 and 37.5 cm. The mean $\bar{h}_{ae,3}$- value is calculated based on all depths.

| Depth | $h_{ae,1}$ | $h_{ae,2}$ | $h_{ae,3}$ |
|---|---|---|---|
| [cm] | [cm] | [cm] | [cm] |
| 2.5 | -0.3 | -3.4 | -108 |
| 7.5 | -1.2 | -4.9 | -107 |
| 12.5 | -2.4 | -4.7 | -104 |
| 17.5 | -1.7 | -9.3 | -96 |
| 22.5 | -1.8 | -10.5 | -72 |
| 32.5 | -11.8* | -31.8 | -117 |
| 37.5 | -10.8* | -20.2 | -132 |
| Mean | -1.5 | -8.0 | -105 |
| SD | -0.7 | -3.2 | -17.1 |





Table 4: Overview is given for different proposed pore domain nomenclature with corresponding pressure head delimitations, and effective pore diameters. * The plant matrix refers to the skeleton of the *Sphagnum* which in its living form contains intact hyaline cells and decomposition of cell walls remaining with the more persistent plant skeleton (Figure 1). **it is not applicable to calculate an effective radius for water

5  which is not contained in capillaries.

| Name of pore domain | Pore domain Number | Effective pore diameter [ μm ] | Pressure head [ cm ] | pF [-] |
|---|---|---|---|---|
| *Inter-plant pores* | I | > 300 | > -10 | 1 |
| *Intra-plant* and *outer plant matrix* pores* | II | 300 to 30 | -10 to -100 | 1 - 2 |
| *Inner-plant* and *plant matrix* pores* | III | 30 to 10 | -100 to -300 | 2 - 2.5 |
| *non-capillary, cytoplasmic and apoplastic water* | IV | na** | <-1000 | > pF 3 |