# Peer review of "A pore-size classification for peat bogs derived from unsaturated hydraulic properties"

_Hydrology and Earth System Sciences, 2017_

## Referee Comment (RC1) · A.J. Baird (Referee) · 24 Jul 2017

Please see my separate report and also a version of the manuscript containing my comments.

Please also note the supplement to this comment:
https://www.hydrol-earth-syst-sci-discuss.net/hess-2017-297/hess-2017-297-RC1-supplement.pdf

---

## Referee Comment (RC2) · Siegel (Referee) · 13 Sep 2017

Review of Tobias K. D. Weber, Sascha C. Iden, and Wolfgang Durner
MS No.: hess-2017-297

Peatland bog pedogenesis is reflected in unsaturated hydraulic
properties

I found this paper interesting with respect to our understanding of how peat degradation changes peat soil properties. In particular, I liked the concept of putting some quantification to peat pore sizes in the acrotelm, the "living" uppr 50 cm or so of peat columns. The shallower the peat, the greater the hydraulic conductivity whereas as the peat becomes more humified hydraulic conductivity—or at least the bulk hydraulic conductivity--becomes less.

Having aid this, I found the authors appear to have missed quite a bit literature on peatland hydrology. Decades of research on peatlands in northern Minnesota and in Canada show that ombrotrophic bogs are not necessarily supported by only precipitation. Clymos, Ingram's and Boelter's early work are dated related to peat soil hydraulic properties. Groundwater discharge can force water upwards into the base of domed bogs, supporting them during drought. To find out if groundwater discharge from regional flow systems provides hydrologic underpinning of bogs, hydrologists now sample pore water into the catotelmic (highly humified peat) and measure at least its pH and specific conductance. The authors might do a search on papers by Paul Glaser, Jeff Chanton, and myself (Donald Siegel). Paul's and my work showed how the hydraulic conductivity of peat may not change with increasing bulk density because of the secondary porosity issue.

These studies and others also show that much of the water that moves through peat can occur through preferential flow paths along fibrous roots and the like, far down into the catotelm. This funneling of water makes modeling the actrotelm using the Richard's equation questionable except at the largest of scales (big areas). Doing experiments on cores won't necessarily capture the system processes. I note that the authors of this paper describe their bog forming in a valley, which would suggest that upward groundwater discharge into it could play a role—not just lateral which they show. And, if groundwater does enter at the bottom of peat profiles, it can dilate the peat pores opening them up and increasing the hydraulic conductivity (D. Ours and Siegel). I see that the authors cite others who have worked on the same bog before, and I recommend they write a few paragraphs describing what others found and from what approach.

I don't have the time to review all the other papers. It could very well be, that this bog in question formed from lake pedogenesis. In this case, the preferential flow issue may not be as important as in the vast mires I and others have studied. The deep peat in paludafied lakes largely comes from decomposing algal remains to my understanding, superceded on top by mosses eventually. So the bog may look like a bog in say, Siberia, but hydraulically it might behave very differently. Paludified catotelmic peat really gets dense and humified compared to peat in other settings and may behave more in line with the Dickey Clymo model. I think the authors need to address this question head on. What kind of bog it is and how did it form?

Ironically, the same problems with the Richardson equation applies to mineral soils in forested terrains. Check out Jeff McDonnel's and Keith Bevin's recent papers on this issue. They argue that new theory has to be developed to address how water moves through unsaturated soils of all types, at least at the scale short of regional synthesis. Indeed, preferential flow is so important that Zeno Levy recently published a paper showing that solutes can move from active pore spaces into dead

pores to create strange geochemical signals in the catotelm (a dual porosity issue). Chanton's isotopic work showed that much of the methane generated in wetlands may come from modern labile carbon that is driven down deep though the acrotelm into the catotelm. Bacteria use this labile carbon to form methane at depth and the methane episodically blows out when the water table drops, and in the process, no doubt creates preferential flow paths all the way to the top.

So, no—I don't buy into the notion that mosses only take water from the upper few cm of peat soils. This makes no sense from what I know of peatland hydrology.

I fully understand that the authors of this paper only looked at the very fibrous peat in the upper 50 cm of the peat column. And that's ok. But they neglect preferential flow and I think this makes their work less useful. The experimental method also strikes me as problematic. In all the years I have taken cores of peat, we take care NOT to freeze them. How can freezing not affect porosity of peat, since frozen water notoriously creates porosity though expansion. Plant cells burst. I see this every time I accidently leave an orange in my freezer too long (for use in drinks). I think the authors need to address this more, rather than cite a paper saying there is no problem.

I also have a problem treating peat porosity akin to soil porosity, talking about pore diameters as if they were spheres. They are not. Peat pores can be linear and vertical, horizontal etc. We are looking at a living system of plants in laid down decomposed plants. Perhaps at the end, some kind of uniform matrix pore distribution can occur, but even then, I see the preferential issue, so clear from a great deal of independent research. At least in mire peat.

Finally, peat can have lower saturated hydraulic conductivity in natural setting because of methane buildup in the pores of question. In natural peatlands, methane can fill up to 20% of lower acrotelmic and then deeper catotelmic peat at times. Check out a paper by Don Rosenberry on this. So flushing out small pieces of peat with DI or air destroys some of the process you want to figure out.

While I have some issues with this paper, I do find some good things too. I liked the hypotheses being proposed and the math seems solid to me. The math results agree with their conclusions, but of course, they would have given thT inversely fitting a model using the Richards equation can be done by uaing different suites of assumed parameterizations. The model is non-unique—as are all mathematical models of hydrology. Even given the lack of recognition for dual porosity and preferential flow being not incorporated into the equation and van Gnutchen solution, I would want to see more sensitivity analysis to show that the parameterization chosen was the best of the lot.

The authors do mention macropores just before their conclusion:

"*that from a soil hydrological perspective, the water storage domain consist of an active and inactive porosity delimited at a pressure head of $h$ = −100 cm".*

While I'm uncertain the -100 cm head applied for peatlands in general, this result needs to be stated or addressed more clearly before the conclusion proper. I think that there may be more active porosity than what the authors think because of methane ebullition and plants other than sphagnum with roots penetrating deeper than 30 cm or so. Are there spruce or other trees in the peat system studied? In places where fen vegetation occurs, I have seen sedge roots go down far deeper, meters. These can funnel labile carbon to depth among other things.

So, my recommendation is for the authors to clarify more for the readers (e.g. when citing a fact like freezing doesn't matter--say why), and address some of the issues I bring up.

I recall a conversation I had with Sandy Verry, who with Boelter, did the earliest work on peat permeability from a small kettle lake bog setting. Paul Glaser and I were coming up with orders of magnitude greater values using piezometers and other sampling devices in the Glacial Lake Agassiz peatland. I asked Sandy what he thought and he said he was not surprised, that they only took samples where they saw no roots and other structures. He said his work reflected matrix and not bulk peat.

Perhaps all that is needed may be for the authors of this paper to state something like that. Recognize that preferential flow may be the driver at larger scales but in its absence, their conceptual model works.

Donald Siegel
Syracuse University

---

## Author Comment (AC1) · 10 Oct 2017

**Reviewer 1 (Andy Baird)**

Dear Andy Baird, we thank you very much for your thorough review of our paper, which was very valuable for improving it. Specifically, we appreciate your a) knowledgeable statements on the phenological differences of *Sphagnum* species and b) your advices with respect to the terminology and hydrological functioning of peatland ecosystems. Moreover, we thank you for the careful attention to detail regarding methodology and the presentation of our manuscript. We have addressed and incorporated your comments in our revised manuscript, as indicated, point by point, detailed below. We have taken great care to make the manuscript more accessible to a broader audience.

**Reviewers Substantive comments:**

Title and use of word 'pedogenesis'
I am not comfortable with the use of the term 'pedogenesis'. The authors use the term to describe physical changes in a peat soil as litter decays and decomposes. Such use of the term would then imply that fresh litter or even partially-decayed litter ('fresh peat') is not true soil. I recommend changing the title of the paper and using a different term to describe peat decomposition and physical alteration.

> We fully agree; the term 'pedogenesis' in the title is potentially misleading. To address this, we have changed the title to:
>
> > "*A pore-size classification for peat bogs derived from unsaturated hydraulic properties*"
>
> , which we think now more directly reflects the adopted methods and obtained results.
>
> The use of the word pedogenesis is now restricted to the abstract and the conclusions of the original manuscript. Since you raise a valid point concerning the applicability of the terminology to *Sphagnum* peat bogs, we have now added the following sentence to the introduction:
>
> > "*In summary, the aforementioned processes constitute the entire continuous pedogenesis shaping the soil profile as an ongoing process (Blume et al., 2016). It should be noted that this includes the actively growing Sphagnum mosses, as Weber et al. (2017a) define the actively growing and living part of the Sphagnum mosses as part of the vadose zone, i.e. as part of the soil profile.*", now on P2 L14-18.
>
> This specification is given directly after on P2 L3-13 where we present with great detail the principle pedogenic processes occurring in Sphagnum bogs is mentioned.
>
> For this, we added the following reference to the reference list:
> Blume, H.-P., Brümmer, G.W., Fleige, H., Horn, R., Kandeler, E., Kögel-Knabner, I., Kretzschmar, R., Stahr, K., Wilke, B.-M.: Scheffer/Schachtschabel Soil Science, Springer Berlin Heidelberg, 618 pp., 2016, doi: 10.1007/978-3-642-30942-7.

Wider applicability of the results vis-à-vis different Sphagnum ssp found in peat bogs.
> This is an important and valid interjection. First, we did not observe any *S. cuspidatum* in the samples from the Odersprungmoor bog on which we conducted our research on. However, in Fig. 1, the images Fig. 1b and Fig 1c do show *Sphagnum cuspidatum*. The intention of the

available images was to exemplify the general architecture of Sphagnum mosses. To address this, we now specify this in the caption of Fig. 1 which reads now:

> *Sphagnum moss structures and soil pore sizes. a) Sphagnum lawn with visible bleaching due to desiccation of the capitula (in German language Sphagnum is also referred to as 'Bleichmoos', which translates to 'bleaching moss'), a-b) images of Sphagnum cuspidatum H.Klinggr to exemplify the b) sampled and slightly spread out individuals with visible inter-connectedness of branches, […].*

Further, we follow your suggestion on potential limitations of our results in the light of different in situ phenologies different species might have. For this, the final paragraph of Section 4 "Proposal of a pore size classification for *Sphagnum* moss and peat" now reads:

> *A word of caution with respect to generalizing our findings: Michaelis (2011) describes 286 species of Sphagnum, occurring globally; thus, our results might not be applicable to all species, since the phenology of decaying Sphagnum might be different between species. Nevertheless, in line with these definitions, a pressure head delimitation of pore water into an active (inter- and intra-plant and inter-plant matrix pore space) in an inactive porosity (inner-plant and inner plant matrix) at a pressure head of h = -100 cm is suggested.*

We have now added Michaelis (2011) to the reference list as:

Michaelis, D.: Die Sphagnum-Arten der Welt, Bibliotheca Botanica, Vol. 160, 408 p. (in German). 2011.

Shrinkage of the samples?
The authors do not say whether their peat samples contracted as they dried. Contraction can have quite substantial effects on the pore-size distribution and will, therefore, affect the fit of any model. It would be useful if the authors could comment on this effect in a revision of the paper. It is widely known that acrotelm peat contracts on drying and it is possible that the small samples used in the study were prevented from shrinking because of friction with the walls of the containers in which they were housed. If so, the results obtained may, to some degree, reflect an artefact of the laboratory setup rather than what happens in the field.

> The effects of shrinkage certainly deserve further research; however, an in depth treatment is considered outside of the scope of this study, since it involves simultaneously simulating transient water fluxes and the change in sample volume.

> To the case in point: First, we would like to point out that Weber et al. (2017) observed volume changes of less than 5-8% referenced to the initial volume for samples from the same bog and the same experimental setup. The experimental setup was reported by Weber et al. (2017) to consist of very thin and flexible latex membranes ensuring a snug fit to the sample, even during drying. We refrain from repeating all the methods which we explicitly addressed in the previous paper.

> Secondly, a large effect cannot be expected as this small change will not affect the presented delimitations which differ in orders of magnitude of pressure head and related effective pore radii. In particular, the results delimiting the larger pores from the hyaline cells at pF = 2 is very distinct which can be seen in Fig. 6 of the manuscript

However, we are currently preparing a manuscript on a study addressing this topic and the preliminary results do not support a significant effect. In particular, the distinct and sharp differentiation between the larger pores and the hyaline cells at pF = 2 are unaffected by this.

To address the detailed statements above we summarise this by adding the following sentence to results section 4 on now P10 L23-26:

*"In our analysis we assume that shrinkage does not affect our key findings. Shrinkage was observed to be around 5-8% on samples from the same depths and same bog, as referenced to the initial volume. Since the delimitations of our pore size classification span orders of magnitude in pressure head and related effective pore radii, we believe that shrinkage will not have a considerable influence on the derived soil hydraulic properties for the small scale"*

**Minor comments**

Variables and parameters in the equations and text
Throughout the paper there is inconsistency in the italicization of variables and parameters. Typically these should be italicized in both the equations and the text (including labels in figures). Regardless of the convention used, the same form should be used in the equations, main body of the text, and figures.

We apologise for this circumstance and have now corrected it in all instances in the equations and text. We note the changes in the revised manuscript through the "track changes" option.

References
This should not happen. We have now double checked the consistency of the reference list and have made changes where applicable, but do not give the details here. We note the changes in the revised manuscript through the "track changes" option.

Minor comments from the comments of the pdf supplementary are listed in the table below:

| Page(s) and Line(s) | Reviewer comment | Reply |
|---|---|---|
| P1,L18 | "Pedogenesis" I am not sure this term is used correctly in the paper. Pedogenesis is the formation of soil from non-soil. In what way is the litter in the acrotelm non-soil? | We have made alterations to address this point in our reply to the major comments (see above). |
| P1,L21 | "Peatland development" Not true for tropical peatlands. I suggest being more specific. | The sentence now specifies that this is the case for temperate and boreal peatbogs. |
| P1, L22 | "Subsequent ombrotrophication": Okay, but in many bogs ombrotrophication occurs first and then Sphagnum establishes. In UK bogs, for example, it is common to see an initial dominance by Eriophorum spp. after the fen-bog transition. | We have corrected the sentences to: "[…], growth of peat bogs and may lead to a manifestation of the ombrotrophication process (Balyea, 2009; Rydin and Jeglum, 2016)." We added the reference Balyea (2009) to the reference list: Balyea, L.R.: Nonlinear Dynamics of Peatlands and Potential Feedbacks on the Climate System, in: (eds) |

| | | Baird, A. J., Belyea, L.R., Comas, X., et al..: Carbon Cycling in Northern Peatlands, Geophysical Monograph 184, Geophysical Monograph Series, American Geophysical Union, 2009. |
|---|---|---|
| P2, L3 | "while SHPs accounting for these processes have only recently been identified by Weber et al. (2017a) for a limited number of samples." It is not quite clear what is meant here by 'SHPs'. The hydraulic properties of the acrotelm have been investigated by a number of authors. I suggest rewording so that the intended meaning is more evident. | We have added the word 'processes' to the first part of the sentence, too. This should clarify, that the second mentioning of 'these processes' points at the water, film, and vapour flow processes for which Weber et al. (2017) identify SHPs. The sentence now reads (changes in italics): "The importance of capillary, film and vapour flow *processes* for upward water fluxes in moss and peat has been emphasized by Hayward and Clymo (1982) and Price et al. (2009) while SHPs accounting for these processes have only recently been identified by Weber et al. (2017a) for a limited number of samples". |
| P2, L4 | "Heavily decomposed" This is often the case, but layers of poorly-decomposed Sphagnum peat may be found throughout a peat profile. The classic acrotelm-catotelm model rarely applies fully to a peat profile and has recently been criticised in the recent scientific literature. | While we are aware of the oversimplification of this continuous change, we first introduce it as a general rule to ease the explanation of the conceptual understanding. At the end of the paragraph we explicitly state cases which cause perturbations to this gradual and smooth decrease: "Exceptions from this rule have been observed in cases where pipe flow (Holden, 2005), fire disturbances (Sherwood et al, 2013), and rapid climate change resulting in changes in vegetation and subsequent peat deposition history (Rydin and Jeglum, 2016, Hedwall et al., 2017) occur". Therefore, we do not see the necessity to change the manuscript. |
| P2, L17 | "Parametrizsation" | We corrected all instances of the incorrect spelling. |
| P2, L17-18 | "SHPs" vs "SHP" | We now SHP for soil hydraulic properties, regardless whether we address it in plural or singular. |
| P2, L17-18 | Meaning of SHPs | It is defined on P1 L28 of the revised manuscript. |
| P2, L21-25 | Ks as predictor and its depth relationship and position of first mentioning of the water retention | The paragraph starting with "However, [...] " until the end of the |

| | | curve and hydraulic conductivity curve in the manuscript. | respective paragraph indeed lacks clarity. We changed it to:

*"While correlations of saturated hydraulic properties, i.e. Ks with depth have been inferred for the upper bog layers of up to 50 cm (Morris et al. 2015) this is not a sufficient predictor for the SHPs. Moreover, knowledge on the pore size density is required to effectively describe the water retention curve (WRC) and the hydraulic conductivity curve (HCC)."* |
|---|---|---|---|
| P2, L24-25 | Introducing WRC and HCC earlier in the text | We think that introducing the abbreviations at this early place in the paper (page 2) is ok. |
| P2, L 34 | PSD mentioned for the first time. | On first mentioning, PSD is now defined as "pore-size distribution. |
| P3, L11 | Widening of PSD with depth | Well spotted. Of course, the reverse is true.

We replaced "widening" with "narrowing". |
| P3, L13 | Is pedogenic the right use of word? | Thank you for double checking on this unusual word. Yes it is the correct word, in so far as it is the adjective of the word pedogenesis. |
| P3, L16 | Wording of: "the research aims of this study are | 'the research aims of the study reported in this paper were'. |
| P3, L21 | Suggestion of deleting "as process model" | We have decided not to follow the suggestions, as we do like the explicit differentiation between the process model, the models for the soil hydraulic properties, and the likelihood model. |
| P3, L26 | Definition of "soli-ombrotrophic" | We learnt that "soli-" is a German prefix, but have now deleted it. The next comment gives greater detail to the hydrology of the peatland. |
| P3, L27 | Minerotrophic influence and valley position (cf als reply to reviewer 2) | We are a little surprised to see that the description is misleading, since Figure 2 (top) very clearly shows the isolines, such that it is clear, that the sampling location is from the ombrotrophic part of the bog. However, since Reviewer 2 has also addressed this concern, we address it by describing the situation differently. |

| | | We changed the sentence on P3, L27-28 of the original manuscript to: *"The Odersprungmoor formed on a saddle with an average downslope of 3 % in the SE-NW direction. In the SW-NE direction it is located in a gentle trough position (Fig. 2; Jensen, 1990)."* Additionally, we state more clearly the results of the hydrology of the very small minerotrophic influence by Broder and Biester (2015) and add more specifically: *"The Odersprungmoor shows features of a poor-fen in some small areas where it is slightly influenced by of minerotrophic water which only occurs on a small strip on the North-Western flank (indicated by the arrows, Fig 1, bottom). Most of the incoming water from the shallow soils in the North-East is diverted past the bog along the northern rim of the bog towards the North-West (Border and Biester, 2015); thus our sampling location is situated in the ombrotrophic part of the bog. Broder and Biester (2015) provide information on the geochemical composition of the substrate and pore waters which supports this.".* |
| P3, L29 | "In shape" --> "in shape in plan" | Done |
| P3, L31 | Give botanical authorities after first use of each Latin name? Also, the pictures from the site in Figure 1 suggest the main Sphagnum species was the terrestrial form of Sphagnum cuspidatum. | Done |
| P3, L31 | On the existence of "*S. cuspidatum*" | We address this in the comment to the caption of Fig. 1. |
| P4, L1 | "Minerotropic water flow into the bog on the North-Western flank" If there is minerotrophic water flowing into the peatland, it is more properly called a fen or 'poor fen'. | cf reply to the comment on P3, L26 and P3, L27. |
| P4, L2 | "On geochemical" → "on the geochemical" | Added 'the' |
| P4, L5 | "Nearby" → "at nearby" | Corrected. |

| P4, L6 | Weak grouping | *"In the acrotelm, a profile characterization with depth is possible according to a weak grouping:"*

Was replaced by
*"In the acrotelm, a profile characterization with depth is possible as follows:"* |
|---|---|---|
| P4, L8 | Humification (von Post) | We do not have this information, and thus refrain from following this suggestion, since we do mention the state of decomposition and give reference to a detailed studies on the geochemical signature of the soil parent material. |
| P4, L11 | 5x → n = 5, … | Corrected as suggested |
| P4, L19 | Explicit introduction of the experimental details | We refrain from giving additional details to make our paper concise. Also, the evaporation experiment is a standard measurement technique in soil physics, and the modifications which were particularly adopted to account for Sphagnum moss has been given careful detail here. However, we added the basic information: *Subsequently, transient evaporation experiments were carried out (Wendroth et al., 1993, Schwärzel et al. 2006) on samples 5 cm i.h. and 8 cm i.d., starting with full saturated samples that were exposed to free evaporation in the lab. Matric potentials were measured in two depths, and water fluxes were derived from weight changes with time*. |
| P5, L22 | "Pressure heads"
How were pressure heads measured? If tensiometers were used, were problems encountered with the contact between the Sphagnum peat and the tensiometer cup? | Yes, with tensiometers, and no loss in contact was observed. |
| P4, L25 | "Model" → models | done! |
| P5, L19 | "estimation circumvents the need to weight the data groups of $\theta(h)$ and log10 K($h$)." Unclear (to me) what is meant here. Perhaps add a line or two of explanation? | To make it clearer, the sentence now reads:
*"The sequential parameter estimation circumvents the need to weight the data groups of $\theta(h)$ and $log_{10} K(h)$, whereas if measured WRC and HCC data are used to* |

| | | *estimate SHP model parameters simultaneously, it involves a weighted multi-objective problem."* |
|---|---|---|
| P5, L26 | "Gravitational acceleration" → acceleration due to free fall | We have not changed this. To our understanding, the used terminology is correct. |
| P5, L26 | L T$^{-1}$ → L T$^{-2}$ | Thank you! We changed that. |
| P6, L12 | 'by the shape' - add 'the' | Done |
| P6, L21 | 'pressure' – singular | Done |
| P6, L21 | „this this" Repeated word. | Done |
| P7, L1 | When not reporting data values, write numbers of nine or less as words? Later in this sentence 'three' and 'one' are used instead of '3' and '1'. | Correct! Changed 4 to four |
| P7, L16 | Choice of word: "Amazingly" | Changed to "very" (reducing the sentence to a factual statement) |
| P7, L24 | Abbreviation of "MSO": What is this? It doesn't appear to have been previously defined. | MSO – multi-step outflow experiments. We have now given the full description of MSO and deleted the abbreviation. |
| P8, L25 | "On the discussion of desiccation tolerance" (There is another way of 'reading' this. If water can be readily lost upwards, we might say the peatland (as opposed to the upper layers of peat) show *less* desiccation tolerance. A peatland that was desiccation-tolerant might be regarded as one where drying of surface peat leads to a hydraulic 'break' (or sharp increase in hydraulic resistance) so that less water is lost from the peatland to the atmosphere. It may be worth adding some more detail/explanation here.) | Since the capitula is exposed to the atmosphere with at times great vapour pressure deficits, evaporation will be an ongoing process. For the capitula not to dry out and get damaged, it is required that a certain pressure head is maintained. This can, in the absence of meteoric water, only be achieved by 1D water flow, vertically upward. To clarify, what we mean, we have now added the word "capitula". The sentence now reads (changes in italics): *"The relatively high hydraulic conductivity in the pressure head range until pF = 2.5 ensures an upward flow of water to the capitula which contributes to the effective desiccation tolerance of the vegetation under field conditions."* |
| P8, L15 | Space between "(" and "Table" | Done. |
| P8, L20 | expression "Carve out" | We like it, too! We have decided to leave it in. |
| P8, L30 | 'This contrasts with reports' - add 'with | done |
| P9, L9 | Pedogenic - See my earlier comments on the use of this term | See detailed comments in the specific section, where we state that |

| | | we now give a definition of the word "pedogenic" and use it |
|---|---|---|
| P9, L8-9 | I think what is written here is reasonable. However, I'm not sure this conclusion will apply in the same way to all the types of Sphagnum peat that can make up the acrotelm. See my separate report. | We treated this in great depth in the specific comments sections. |
| P10, L1 | 'exist' not 'exists' | done |
| P10, L10 | 'skeletal' – spelling | done |
| P10, L14 | This should say 'and'. | done |
| P10, L23 | Comma needed here. | done |
| P10, L25 | A semi-colon or full stop (period) is needed here rather than a comma. | done |
| P10, L30 | This should be 'of'. | done |
| P10, L31 | Italics needed. | done |
| P11, L5 | Delete „to" | done |
| P11, L13 | "under different boundary conditions" delete" | done |
| P11, L14 | Rations → ratios | done |
| P11, L15 | Ks predictions from C/N ratios: I don't think this is true. Saturated K does not show strong relationships with the listed metrics. If you disagree please add some supporting references. | Deleted "*as is often done for saturated conductivity*", since, on double checking the literature, the statement is ambiguous. |
| P11,L17 | Code and data availability | Sorry, but while we embrace the notion of open access (since we submitted this article to HESSD), we cannot do this with the current codes. However, we have uploaded the HYDRUS project files which contain all necessary data of the evaporation experiments and state this in the Data and code availability section. |
| References and acknowledgement | | Done |
| Figure 1 | S. cuspidatum vs S. magellanicum | Cf specific comments |
| Figure 2 | Dimension and reference to elevation | |
| Figure 3 | Italics "K" | Done |
| Figure 4 | Suggestion to use y label only once. | We opted to repeat the y axis label in each row enabling other researchers to use some of the subplots for their own use in e.g. teaching. We did not change this. |
| Figure 5 | Italics "K" | Done |

| | | |
|---|---|---|
| Figure 6 | subscript | Done, we have now added a second x axis at the top of the graph with the effective pore diameters |
| Figure 7 | Different font sizes | Done |
| Figure 8 | Different font sizes | This was intentional due to the long second x-axis label |
| Table 1, P26, L4-7 | This is difficult to read. I suggest breaking it up into a couple of sentences or improving the punctuation. Also, what does 'L' refer to when mentioning heights? | Thank you for this sensible suggestion. We have reorganized the caption. It now reads: Table 1: Statistical evaluation results of the inverse parameter estimation for 31 samples of eight mid sampling depths. The definitions of the abbreviations are given as footnotes to the table |
| Table 2 | Italics needed? | No. Was corrected. |
| Table 3 | Font difference | We trust this will be sorted out in the typesetting, but MS Word gives us no indication on font size differences. |
| Table 4 | Is it needed? | Yes. We think summarising this classification is very helpful to quickly capture the essence of the paper, as e.g. also Hayward and Clymo (1982) did when they presented their seminal work with results not unlike ours. |

References used.

Hayward, P. M. and Clymo R. S.: Profiles of Water Content and Pore Size in Sphagnum and Peat, and their Relation to Peat Bog Ecology, Proc. Roy. Soc. B. Bio., 215(1200), 299-325, doi:10.1098/rspb.1982.0044, 1982.

---

## Author Comment (AC2) · 10 Oct 2017

**Reviewer 2 (Daniel Siegel)**

Dear Dr. Siegel, we thank you very much for taking the time and effort to review our paper. You have provided some very interesting reflections upon the greater ecosystem and hydrology within which the acrotelm and the specific sampling location is situated and its potential influence upon our conclusions. In spite of your closing comments "*Perhaps all that is needed may be for the authors of this paper to state something like that [preferential flow]. Recognize that preferential flow may be the driver at larger scales but in its absence, their conceptual model works*", we have answered each of your topical paragraphs below in a sequential manner. However, we would like to point out that our improvements on the manuscript have not lead to fundamental changes in our manuscript, because this would completely shift the focus from a detailed and intense study on the (unsaturated) soil hydraulic characteristics and property functions of the investigated peatland profile to a general study about peatland hydrology on a large scale, including preferential water flow and transport and its effect on pore water composition. This was clearly not the intention of or paper, and we hope that with the changes, we have made it clearer now.

As a general point, we would like to stress that (as also addressed in the reply to reviewer 1) we apparently used an originally ambivalent description of the peatland situation in our paper. To improve this, we have amended the first paragraphs of the Material and Methods section which now reads:

> "*The samples were collected at an ombrotrophic peat bog, the Odersprungmoor, Harz Mountains, Central Germany (UTM 32U 608000 mE 5737000 mN; 800 to 821 m a.s.l.). The Odersprungmoor formed on a saddle with an average downslope of 3 % in the SE-NW direction. In the SW-NE direction it is located in a gentle trough position (Fig. 2; Jensen, 1990). […]. The Odersprungmoor shows features of a poor-fen in some small areas where it is slightly influenced by minerotrophic water on a small strip on the North-Western flank (indicated by the arrows, Fig 1, bottom). Most of the incoming water from the shallow soils in the North-East is diverted past the bog along the northern rim of the bog towards the North-West (Border and Biester, 2015); thus our sampling location is situated in the ombrotrophic part of the bog. Broder and Biester (2015) provided information on the geochemical composition of the substrate and pore waters which supports this.* "*

We hope, the brevity with which we present the current understanding of this particular peatland is more than sufficient to comprehend our study. Futher, we follow your suggestion on a statement on the dominance of preferential flow on the large scale by adding the following statement to the end of the results section 3.3 now on P9 L28-29.

> "*Without contradiction to the presented model, we point out that preferential flow may be an important driver for saturated flow on the very large.*"

P1C1: Having said this, I found the authors appear to have missed quite a bit literature on peatland hydrology. Decades of research on peatlands in northern Minnesota and in Canada show that ombrotrophic bogs are not necessarily supported by only precipitation. Clymos, Ingram's and Boelter's early work are dated related to peat soil hydraulic properties. Groundwater discharge can force water upwards into the base of domed bogs, supporting them during drought. To find out if groundwater discharge from regional flow systems provides hydrologic underpinning of bogs, hydrologists now sample pore water into the catotelmic (highly humified peat) and measure at least its pH and specific conductance. The authors might do a search on papers by Paul Glaser, Jeff Chanton, and myself (Donald Siegel). Paul's and my work showed how the hydraulic conductivity of peat may not change with increasing bulk density because of the secondary porosity issue.

P2C1: These studies and others also show that much of the water that moves through peat can occur through preferential flow paths along fibrous roots and the like, far down into the catotelm. This funneling of water makes modeling the actrotelm using the Richard's equation questionable except at the largest of scales (big areas). Doing experiments on cores won't necessarily capture the system processes. I note that the authors of this paper describe their bog forming in a valley, which would suggest that upward groundwater discharge into it could play a role—not just lateral which they show. And, if groundwater does enter at the bottom of peat profiles, it can dilate the peat pores opening them up and increasing the hydraulic conductivity (D. Ours and Siegel). I see that the authors cite others who have worked on the same bog before, and I recommend they write a few paragraphs describing what others found and from what approach.

> The reviewer is certainly right with respect to the general characterization of peatland hydrology and the composition of pore waters in such systems. However, our paper is focused specifically on the characterization of the soil hydraulic properties and their model representation, i.e. models for the water retention and hydraulic conductivity of the variably saturated zone in topmost 40 cm of an ombrotrophic bog. We intentionally do not present our study as a peatland hydrology research.

> We acknowledge that the flow paths in peat bogs might lead to an influence of minerotrophic groundwater, but Border and Biester (2015) do not support this for our site on a hydrogeochemical analyses of pore water from the top part of the saddle (our sampling location). To address this, we have rewritten the description of the site location.

> Considering the relationship between saturated conductivity and bulk density, we agree. For this reason, and also by addressing a respective comment of reviewer 1, we have deleted the sub clause "*as is often done for saturated conductivity*" in the second but last sentence of the conclusions. Moreover, we acknowledge this fact in the rationale of our study, stating that "*the saturated conductivity is not necessarily a good predictor for the unsaturated hydraulic properties*." (now P2 LL24-25). We furthermore now clearly state that this might be an effect macro-pores can have on the saturated conductivity, by adding the following sub clause to the sentence: "*, since the contribution of macro-porosity on preferential flow can be substantial in the saturated case*". (now P2 LL25-26).

P3: I don't have the time to review all the other papers. It could very well be, that this bog in question formed from lake pedogenesis. In this case, the preferential flow issue may not be as important as in the vast mires I and others have studied. The deep peat in paludafied lakes largely comes from decomposing algal remains to my understanding, superceded on top by mosses eventually. So the bog may look like a bog in say, Siberia, but hydraulically it might behave very differently. Paludified catotelmic peat really gets dense and humified compared to peat in other settings and may behave more in line with the Dickey Clymo model. I think the authors need to address this question head on. What kind of bog it is and how did it form?

> The focus of our work is different. We do not want to give a generalized characterization for a certain type of peatland, but want to show that under the site of our investigation, we can characterize the depth profile of soil hydraulic properties with a smooth and gradual transition from the three-modal porosity properties of the living moss to the bimodal porosity of decomposed peat. Certainly, it would be desirable to assess the generality of this finding with respect to different types of peatland, but this would require investigations at different sites with an associated enormous amount of work. Our paper reports results from a very intense and detailed study at one specific site. How general this finding is, and to what extent it can be applied to similar or other types of peatlands, is not part of the study.

Our site is not and cannot have been formed from lake pedogenesis, since it is situated on a saddle with a watershed divide at it's crescent. We have addressed this point in a more detailed manner by adding quite a bit of explanation to the Site description in section 2. From this, we hope to clarify this issue. There is already quite a bit of description in P3 L25-P32 of the original manuscript and from the isolines in Figure 2. We have also addressed this in the comments to reviewer 1 by making the site description less ambivalent. Sadly, there is no information in the literature on genesis of the Odersprungmoor, and peat coring and subsequent description is not the scope of this paper. We do acknowledge, that a study on this presents itself as a very interesting research question.

P4: Ironically, the same problems with the Richardson equation applies to mineral soils in forested terrains. Check out Jeff McDonnel's and Keith Bevin's recent papers on this issue. They argue that new theory has to be developed to address how water moves through unsaturated soils of all types, at least at the scale short of regional synthesis. Indeed, preferential flow is so important that Zeno Levy recently published a paper showing that solutes can move from active pore spaces into dead pores to create strange geochemical signals in the catotelm (a dual porosity issue). Chanton's isotopic work showed that much of the methane generated in wetlands may come from modern labile carbon that is driven down deep though the acrotelm into the catotelm. Bacteria use this labile carbon to form methane at depth and the methane episodically blows out when the water table drops, and in the process, no doubt creates preferential flow paths all the way to the top.

So, no—I don't buy into the notion that mosses only take water from the upper few cm of peat soils. This makes no sense from what I know of peatland hydrology.

To the best of our knowledge, we do not state that mosses only take water from the upper few cm of peat soils. We describe the uppermost part of an ombrotrophic peatland, in its transition from living moss to decayed peat, as a porous system that can be successfully characterized with the framework of the classical continuum description of water flow in porous media with hydraulic properties that gradually change with depth. We like to stress that this result is new with respect to detail of the characterized pore systems and representative for the local situation. The specific hydraulic dynamics in this system will of course depend on the boundary conditions, and water flow will occur upwards and downwards under transient boundary conditions.

We do not extend our study to a hydrological modelling of the whole catchment.

We are familiar and aware of the solute transport properties of porous media exhibiting preferential flow paths. But since we are looking at soil matrix water storage and conductivity, we think that our types of experiments are well thought out and designed.

I fully understand that the authors of this paper only looked at the very fibrous peat in the upper 50 cm of the peat column. And that's ok. But they neglect preferential flow and I think this makes their work less useful. The experimental method also strikes me as problematic. In all the years I have taken cores of peat, we take care NOT to freeze them. How can freezing not affect porosity of peat, since frozen water notoriously creates porosity though expansion. Plant cells burst. I see this every time I accidently leave an orange in my freezer too long (for use in drinks). I think the authors need to address this more, rather than cite a paper saying there is no problem.

We do not see any problem, here. Of course, freezing might have an effect on even unsaturated samples. However, the *Sphagnum* moss and peat we used freezes every winter in the field. So we are treating it in a way nature does every winter anyway. In terms of the

method itself, it is quite common to freeze peat bog samples (just one of the many examples is McCarter and Price, 2012).

I also have a problem treating peat porosity akin to soil porosity, talking about pore diameters as if they were spheres. They are not. Peat pores can be linear and vertical, horizontal etc. We are looking at a living system of plants in laid down decomposed plants. Perhaps at the end, some kind of uniform matrix pore distribution can occur, but even then, I see the preferential issue, so clear from a great deal of independent research. At least in mire peat.

> We fully agree on your thoughtful comments on the size and orientation of the soil pores. Please note that soil hydraulic properties, also for mineral soils, always characterize **effective** properties. Any direct interpretation of the corresponding pore-size distributions as direct capillary diameters would be wrong. However, the interpretation of "pore size" just by the energetic state of pore water is common practice in soil physics, and we do not see any necessity to differentiate between peat and mineral soils in this respect.

Finally, peat can have lower saturated hydraulic conductivity in natural setting because of methane buildup in the pores of question. In natural peatlands, methane can fill up to 20% of lower acrotelmic and then deeper catotelmic peat at times. Check out a paper by Don Rosenberry on this. So flushing out small pieces of peat with DI or air destroys some of the process you want to figure out.

> Please note that we characterize her ethe effective pore-size distribuition of the investigated peat soil as material property. Yes, entrapped gases block largest pores and thus can greatly reduce the permeability for the water. Therefore, in a hydraulic modeling of water dynamics under natural conditions, the possible formation of gas must be explicitly considered. However, again we stress that we give in this paper a sound soil physical base for hydrologic simulations. Whether these include methane buildup or not, would be dependent on the purpose of the related hydrological modeling. In any case, such an application would require an additional conceptual treatment of the blocking effect by gas formation. In our characterization, we specify just the permeability for the wetting phase.

While I have some issues with this paper, I do find some good things too. I liked the hypotheses being proposed and the math seems solid to me. The math results agree with their conclusions, but of course, they would have given the inversely fitting a model using the Richards equation can be done by using different suites of assumed parameterizations. The model is non-unique—as are all mathematical models of hydrology. Even given the lack of recognition for dual porosity and preferential flow being not incorporated into the equation and van Genutchen solution, I would want to see more sensitivity analysis to show that the parameterization chosen was the best of the lot.

> Thank you for the good words. In contrast to your statement, we did incorporate dual- and tri-modality (which is able to map macroporosity in the framework of the Darcy-Buckingham law into soil hydraulic properties) in both soil hydraulic property model groups. We found that the model parameters are identifiable and unique, we added this by adding
> *"It furthermore corroborates the fact that even model parameters of three-modal functions can be uniquely identified."* to the manuscript on page 7 LL11-12.

While I'm uncertain the -100 cm head applied for peatlands in general, this result needs to be stated or addressed more clearly before the conclusion proper. I think that there may be more active porosity than what the authors think because of methane ebullition and plants other than sphagnum with roots penetrating deeper than 30 cm or so. Are there spruce or other trees in the peat system studied? In places where fen vegetation occurs, I have seen sedge roots go down far deeper, meters. These can funnel labile carbon to depth among other things. So, my recommendation is for the authors to clarify more for the readers (e.g. when citing a fact like freezing doesn't matter--say why), and address some of the issues I bring up.

> We do not study any parts of the poor-fen peatland, and yes, some very small and few spruces can be found, while these have a potential large impact on saturated flow and solute transport of the larger hydrology, this cannot be expected of the unsaturated characteristics.

I recall a conversation I had with Sandy Verry, who with Boelter, did the earliest work on peat permeability from a small kettle lake bog setting. Paul Glaser and I were coming up with orders of magnitude greater values using piezometers and other sampling devices in the Glacial Lake Agassiz peatland. I asked Sandy what he thought and he said he was not surprised, that they only took samples where they saw no roots and other structures. He said his work reflected matrix and not bulk peat. Perhaps all that is needed may be for the authors of this paper to state something like that. Recognize that preferential flow may be the driver at larger scales but in its absence, their conceptual model works.

> We addressed this

References

McCarter, C. P. R. and Price J. S.: Ecohydrology of Sphagnum moss hummocks: mechanisms of capitula water supply and simulated effects of evaporation, Ecohydrol., 7(1):33-44. doi:10.1002/eco.1313, 2012.

Yours Sincerely,

Tobias Weber, Braunschweig, 15 September 2017

--
**Tobias KD Weber (Geoecologist and Soil Hydrologist)**
Researcher in CRC 1253 CAMPOS
University of Hohenheim
Institute of Soil Science and Land Evaluation | Biogeophysics
Emil-Wolff-Straße 27 | D-70593 Stuttgart
tel.: +49 (0) 711 459-24019 (direct) | tel.: +49 (0) 711 459-23765 (secretary)
web: https://www.uni-hohenheim.de/en/organization/institution/biogeop

---

## Author Response (AR1)

Revisions of the manuscript Weber et al. '*Peatland bog pedogenesis is reflected in unsaturated hydraulic properties*' originally submitted to Hydrology and Earth System Sciences Discussions on 19 May 2017

Dear Dr. Brian Berkowitz,

10 thank you for granting us the possibility to revise and improve our manuscript. We have taken the opportunity to carefully revise the manuscript as indicated in our detailed point-by-point replies to the reviewer comments (please refer to the uploaded documents).

Below you will find the version of the manuscript with all revisions included, which we have marked accordingly using the 'track changes' option.

15 Additionally, we have uploaded the revised manuscript without marked changes and the full set of supplementary information, as indicated in our reply to the reviewers.

Yours Sincerely,

20

5

Tobias Weber, Braunschweig, 17 October 2017

-

Tobias KD Weber (Geoecologist and Soil Hydrologist)

25 Researcher in CRC 1253 CAMPOS

University of Hohenheim

Institute of Soil Science and Land Evaluation | Biogeophysics

Emil-Wolff-Straße 27 | D-70593 Stuttgart

tel.: +49 (0) 711 459-24019 (direct) | tel.: +49 (0) 711 459-23765 (secretary)

 $30 \quad web: https://www.uni-hohenheim.de/en/organization/institution/biogeop$

[revised manuscript text omitted]

**Figures**